# Proteolytic processing of palmitoylated Hedgehog peptides specifies the 3-4 intervein region of the *Drosophila* wing

**Sabine Schürmann[1,2], Georg Steffes[3,4], Dominique Manikowski[1,2], Philipp Kastl[1,2], Ursula Malkus[5], Shyam Bandari[1,2], Stefanie Ohlig[1,2], Corinna Ortmann[1,2], Rocio Rebollido-Rios[3], Mandy Otto[1,2], Harald Nüsse[5], Daniel Hoffmann[3], Christian Klämbt[4], Milos Galic[5], Jürgen Klingauf[5], Kay Grobe[1,2]***

[1]Institute of Physiological Chemistry and Pathobiochemistry, University of Münster, Münster, Germany; [2]Cells-in-Motion Cluster of Excellence (EXC1003-CiM), University of Münster, Münster, Germany; [3]Center for Medical Biotechnology, University of Duisburg-Essen, Essen, Germany; [4]Institute of Neurobiology, University of Münster, Münster, Germany; [5]Institute of Medical Physics and Biophysics, University of Münster, Münster, Germany

**Abstract** Cell fate determination during development often requires morphogen transport from producing to distant responding cells. Hedgehog (Hh) morphogens present a challenge to this concept, as all Hhs are synthesized as terminally lipidated molecules that form insoluble clusters at the surface of producing cells. While several proposed Hh transport modes tie directly into these unusual properties, the crucial step of Hh relay from producing cells to receptors on remote responding cells remains unresolved. Using wing development in *Drosophila melanogaster* as a model, we show that Hh relay and direct patterning of the 3–4 intervein region strictly depend on proteolytic removal of lipidated N-terminal membrane anchors. Site-directed modification of the N-terminal Hh processing site selectively eliminated the entire 3–4 intervein region, and additional targeted removal of N-palmitate restored its formation. Hence, palmitoylated membrane anchors restrict morphogen spread until site-specific processing switches membrane-bound Hh into bioactive forms with specific patterning functions.
DOI: https://doi.org/10.7554/eLife.33033.001

*For correspondence:
kgrobe@uni-muenster.de

## Introduction

Hedgehog (Hh) morphogens are dually lipidated 19 kDa proteins that are firmly anchored to the cell membrane of producing cells. Production of all active Hhs begins with autocatalytic cleavage of a precursor molecule by its C-terminal cholesterol transferase domain (*Porter et al., 1996b*). This results in cholesteroylated vertebrate Sonic hedgehog (Shh) and *Drosophila* Hh. Next, Hh acyltransferase (Hhat, also designated Skinny hedgehog or Raspberry) attaches a palmitoyl group to a conserved N-terminal cysteine that becomes exposed after signal peptide cleavage (*Chamoun et al., 2001*; *Lee and Treisman, 2001*; *Micchelli et al., 2002*). Hh palmitoylation is critical for later signaling, demonstrated by mutation of the N-terminal cysteine to serine or alanine (C25 > A/S in Shh$^{C25A/S}$, C85 >A/S in *Drosophila* Hh$^{C85A/S}$) which abolishes palmitoylation and results in morphogen inactivity (*Chamoun et al., 2001*; *Chen et al., 2004*; *Dawber et al., 2005*; *Goetz et al., 2006*; *Kohtz et al., 2001*; *Lee et al., 2001*; *Pepinsky et al., 1998*). However, why N-palmitoylation is required for Hh signaling in vivo is still unclear.

Another unusual feature of all Hhs is their multimerization at the surface of producing cells which requires binding to the long, unbranched heparan sulfate (HS) chains of cell surface HS

**eLife digest** Each cell in a developing embryo receives information that determines what type of body structure it will form. In fruit flies, this information is partly given by a protein called Hedgehog. In the embryo cells that receive it, Hedgehog can trigger a series of events which activate certain genes and thereby regulate structure formation.

The Hedgehog proteins are produced by a different organizing group of cells: from there they transport within the embryo, creating a gradient. Depending on where a responding cell is in the embryo, it receives a different amount of Hedgehog, which gives the cell its identity. For example, Hedgehog proteins form a gradient across a fruit fly's developing wing, which creates a visible vein pattern. How Hedgehog proteins form gradients is enigmatic, however, because once produced, they cling to the cells that created them.

The reason for this unusual behavior is that the two ends of the Hedgehog protein are attached to a different fat molecule. In particular, one extremity is linked to a fat molecule called palmitate. These ends' fatty additions anchor Hedgehog to the cells that produced them. Then, the tethered proteins gather together to form chain-like clusters where they inactivate each other: the extremity with the palmitate 'hides' the portion of the neighboring protein that binds to the receiving cells. It is still unclear how Hedgehog can be activated and released to reach these faraway cells.

One hypothesis is that an enzyme comes to the clusters and frees the proteins by cutting both of Hedgehog's fatty anchors. Thanks to how the palmitate tethers Hedgehog to the cell, the protein is positioned in such a way that when the enzyme makes its snip, the binding site on the neighboring Hedgehog gets exposed: this protein is activated and, when also cut by the enzyme, released.

Here, Schürmann et al. create an array of mutant Hedgehog proteins – for example some without palmitate, some with palmitate that cannot be removed by the enzyme – and study how they affect the development of the wing's pattern in the fruit fly. Coupled with the imaging of the clusters, these experiments support the hypothesis that the palmitate anchor is necessary so that Hedgehog proteins can be turned on before diffusing away.

The Hedgehog family of proteins is also present in humans, where it presides over the development of the embryo but is also involved in cancer. Understanding how Hedgehog works in the fruit fly could lead to new discoveries in humans too.

DOI: https://doi.org/10.7554/eLife.33033.002

proteoglycans (HSPGs) called glypicans (*Chang et al., 2011*; *Ortmann et al., 2015*; *Vyas et al., 2008*). The Hh cholesterol modification is sufficient to drive this process (*Feng et al., 2004*; *Gallet et al., 2006*; *Koleva et al., 2015*; *Ohlig et al., 2011*). Despite membrane anchorage and cell-surface HS association, the multimeric Hhs initiate the Hh response in distant cells that express the Hh receptor Patched (Ptc). The question of how dual-lipidated Hh clusters manage to travel and signal to remote target cells is intensely investigated. The most current models propose lipidated Hh transport on filopodia called cytonemes (*Bischoff et al., 2013*; *Sanders et al., 2013*) or on secreted vesicles called exosomes (*Gradilla et al., 2014*) to bridge the distance between Hh-producing and receiving cells.

Hh release through cell-surface-associated proteases, called sheddases, has also been suggested. In vitro, membrane-proximal shedding not only releases Hh ectodomains from their lipidated N-terminal peptides (*Dierker et al., 2009*; *Ohlig et al., 2011*) but also activates Hh clusters. This is because N-terminal lipidated peptides block adjacent Hh-binding sites for the receptor Ptc and, thereby, render Hh at the cell membrane inactive. By cleaving these inhibitory peptides during release, sheddases unmask Ptc binding sites of solubilized clusters and thereby couple Hh solubilization with its bioactivation. In this model, the N-palmitate plays two indirect roles for Hh biofunction: first, it ensures reliable membrane-proximal positioning of inhibitory N-terminal peptides as a prerequisite for their efficient proteolytic processing, and second, by its continued association with the cell membrane, it ensures that only fully processed (=activated) Hh clusters are released. This model therefore predicts that inhibition of N-palmitoylation will result in release of inactive soluble proteins with masked Ptc-binding sites (*Jakobs et al., 2014*; *Jakobs et al., 2016*; *Ohlig et al., 2011*;

*Ohlig et al., 2012*). It also predicts that impaired or delayed processing of dual-lipidated Hh will strongly reduce its release and bioactivity in vivo.

By uncovering a dominant negative, cell-autonomous function of non-palmitoylated Hh[C85S] in endogenous Hh, we here support the first prediction. By using a series of transgenic *Drosophila melanogaster* lines that express untagged Hh, biologically inactive Hh[C85S], or N-truncated variants thereof in posterior and anterior wing disc compartments, we provide strong evidence that Hh clusters form by direct protein-protein contact and that unprocessed N-terminal peptides block Ptc binding of adjacent endogenous Hhs. As a consequence, we suggest that, due to their reduced activity, soluble clusters with masked Ptc-binding sites impair direct patterning of the 3–4 intervein region of the wing. Supporting this mechanism, targeted deletion on non-palmitoylated inhibitory peptides restores 3–4 intervein formation. We also show that impaired or delayed processing of lipidated Hh strongly reduces its solubilization, and hence its bioactivity, in vivo. We demonstrate that the HS-binding Cardin-Weintraub (CW) motif serves as the preferred N-terminal Hh processing site in vivo, and that impaired processing of this site completely abolishes direct 3–4 intervein wing patterning. Additional targeted deletion of N-palmitate restores wing patterning, demonstrating that one role of palmitoylated membrane anchors is to prevent the release of un- or incompletely processed Hh clusters in vivo. These genetic data are supported by the nano-structure of Hh clusters as revealed by immunoelectron microscopy (IEM) and provide new insights into how Hh relay from the producing cell membrane or between membranes could be achieved.

## Results

### Visualization of Hh multimer nano-architecture by IEM

A first step in decoding possible Hh solubilization modes is to characterize the composition and organization of Hh substrates. It has been previously shown that Hh forms light microscopically visible clusters at the surface of producing cells (*Chen et al., 2004*; *Gallet et al., 2006*; *Ortmann et al., 2015*; *Sanders et al., 2013*; *Vyas et al., 2008*). However, the nanoscale structure of these heteroprotein complexes has not been determined. We therefore expressed Shh together with Hh acyltransferase in HEK293-derived Bosc23 cells to produce authentic cell surface Hh clusters for IEM analysis. To this end, we used several different α-Shh antibodies and secondary antibodies conjugated to 5 nm or 10 nm gold particles. Three α-Shh antibodies detected Shh in variably sized cell surface clusters, with the largest complexes exceeding sizes of 100 nm. Notably, as shown in *Figure 1a–h* and *Figure 1—figure supplement 1*, many clusters consisted of linear arrangements (*Figure 1a–d*) or contained linear arrays of closely packed gold particles (*Figure 1f,g*, arrowheads). Nearest-neighbour analysis of the angular distribution between the three most proximal gold particles (*Figure 1i*) confirmed that most arrangements were rectangular (90°) or linear (180°), the latter being consistent with Hh multimerization using linear HS chains of glypican HSPGs as templates (*Chang et al., 2011*; *Vyas et al., 2008*; *Schuermann et al., 2018*). Hh linearization during cell-surface multimerization is further consistent with previous structural and biochemical data which suggest a zig-zag arrangement of Hh monomers (*Figure 1j*) and variably sized *Drosophila* Hh and vertebrate Shh, ranging from 80 kDa to 600 kDa (*Chang et al., 2011*; *Chen et al., 2004*; *Jakobs et al., 2014*; *Ohlig et al., 2011*; *Ohlig et al., 2012*). We therefore next aimed to genetically confirm direct Hh clustering in vivo by using *Drosophila melanogaster* wing development as a model.

### Multimeric Hh[C85S] inhibits wild-type Hh function

The fly wing develops from the imaginal wing disc (*Figure 2a*, bottom). The wing primordium at the center of the wing disc differentiates into the wing blade proper, which shows a characteristic pattern of five longitudinal veins (L1-5), an anterior cross vein (connecting L3 and L4) and a posterior cross vein (connecting L4 and L5) (*Figure 2a*, top) (*Hartl and Scott, 2014*). The anterior/posterior (a/p) boundary is located slightly anterior to the position of L4 in the adult wing (*Figure 2a*, red dashed line).

Hh is produced in the posterior wing disc compartment under the control of the transcription factor Engrailed (en) (*Tabata et al., 1992*; *Zecca et al., 1995*), which acts indirectly on Hh expression through the repression of the transcriptional Hh repressor Cubitus interruptus (Ci) (*Bejarano and*

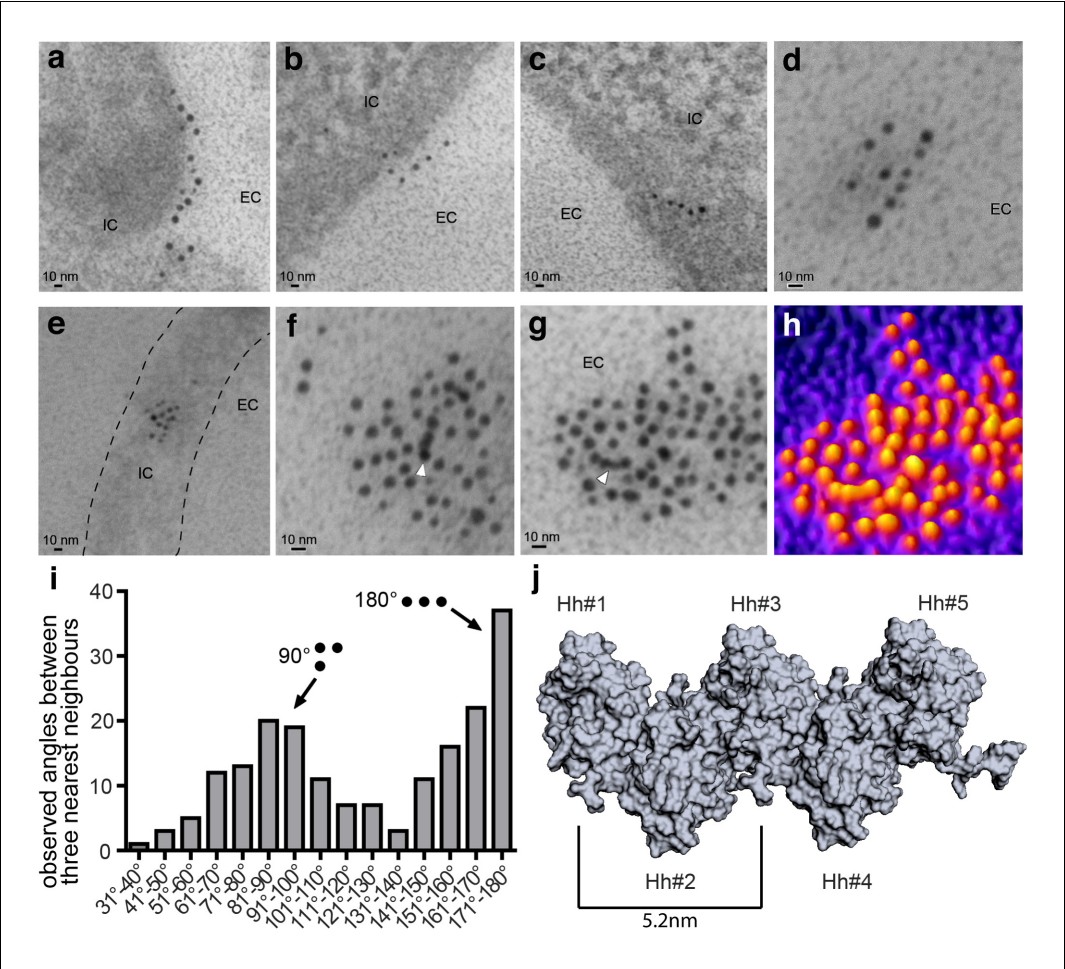

**Figure 1.** Immuno-TEM analysis of Shh clusters at the Bosc23 cell surface (a–g). α-Shh immunogold labeling suggests that Shh forms linear arrangements (EC: extracellular; IC: intracellular). Shh and Hh acyltransferase were coexpressed in Bosc23 cells and cell-surface-associated clusters visualized by two independent α-Shh antibodies and 5 nm and 10 nm immunogold-labeled secondary antibodies. Note Shh clusters on cellular extensions (dashed line in e) and continuous immunogold labeling in cell-surface aggregates (f,g, white arrowheads) consistent with linear Shh clustering. (h) 3D heat map conversion of the cell-surface cluster shown in g. Bright yellow indicates 5 nm and 10 nm gold, and dark colors represent the background. (i) Quantification of angular distributions of the three most adjacent gold particles within clusters confirm non-random Shh clustering. 187 angles in 20 individual cell-surface clusters were analyzed. (j) Hh pentamer model. Hh monomers form extended zigzag chains (*Ohlig et al., 2011*).

DOI: https://doi.org/10.7554/eLife.33033.003

The following figure supplement is available for figure 1:

**Figure supplement 1.** Immuno-TEM examples of Shh clusters at the Bosc23 cell surface.

DOI: https://doi.org/10.7554/eLife.33033.004

*Milán, 2009*). Hh then moves across the a/p boundary into the anterior compartment, where it binds to Ptc (*Ingham et al., 1991*). During its movement, Hh forms a gradient of decreasing concentration with increasing distance from the a/p border which corresponds to differential activation of different Hh target genes. Up to ten cell diameters from the a/p boundary, high Hh levels directly pattern the central L3-L4 region of the wing (*Mullor et al., 1997*; *Strigini and Cohen, 1997*) by stabilizing Ci[155]. More distal regions, up to 12–15 cell diameters from the a/p border, depend on Dpp, which is secreted in a stripe just anterior to the a/p boundary in response to low Hh levels. Hh thus plays a role in *Drosophila* wing patterning by controlling the spatially defined expression of target genes at the a/p border.

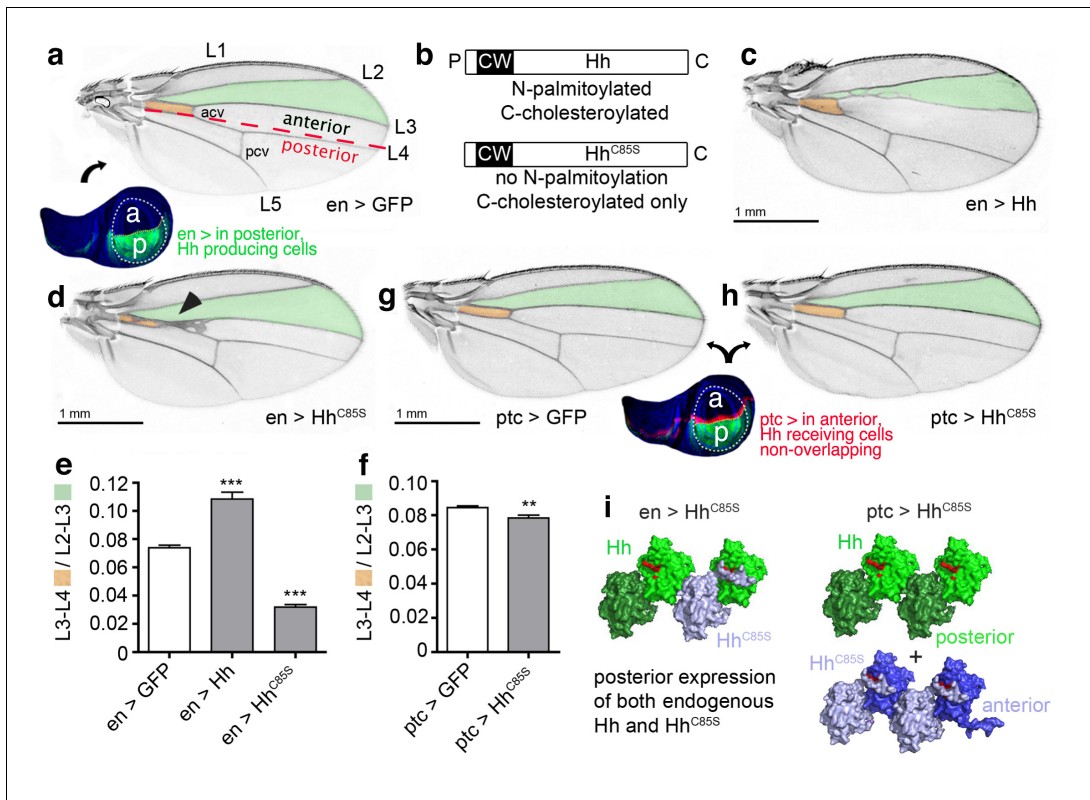

**Figure 2.** Hh[C85S] expressed in the posterior wing disc compartment, but not in the anterior compartment, dominantly represses the formation of Hh-dependent wing structures (**a**) *Drosophila* third-instar wing disc and adult wing. An *en >GFP* control wing disc and wing are shown. The posterior en-expression domain of the wing disc is labeled in green. Adult wings are shown with anterior up and proximal left. Longitudinal veins L1-L5, the anterior cross vein (acv) and the posterior cross vein (pcv) are marked. The anterior (**a**)/posterior (**p**) compartment border is shown as a red dashed line. (**b**) Schematic of transgenic constructs. P: palmitate, C: cholesterol, CW: Cardin-Weintraub motif. (**c**) Hh overexpression under en-control (*en >Hh*) expands the anterior L3-L4 intervein region. (**d**) *en >Hh[C85S]*: L3 and L4 appose proximally; the acv is lost. (**e,f**) To quantify Hh activity, the most proximal L3-L4 areas highlighted in orange were determined and the values obtained divided by the L2-L3 areas. ***p≤0.001, **p≤0.01, n = 20. (**g,h**) *Ptc*-controlled GFP and Hh[C85S] expression in the anterior (Hh-receiving) wing disc compartment at the a/p border (red stripe) do not impair wing development. (**i**) Left: Proposed mixed composition of morphogen clusters upon transgenic Hh[C85S] (gray) expression in the posterior compartment that simultaneously produces the wild-type morphogen (green). Here, the N-terminal peptide of one molecule in the chain blocks the Ptc-receptor-binding site (red) of the adjacent molecule in the chain. Shh crystal lattice interactions (pdb: 3m1n) are shown to illustrate a possible cluster structure. Right: Hh[C85S] expression in the anterior compartment (under ptc control) prevents the assembly of mixed morphogen clusters, leaving endogenous Hh function unimpaired.

DOI: https://doi.org/10.7554/eLife.33033.005

We exploited the Hh-regulated wing patterning response as a simple and reliable in vivo assay to test the functional consequences of proteolytic Hh processing. Specifically, we addressed the formation and positioning of longitudinal L3-L4 veins, and investigated whether Hh proteolytic processing in cells of the posterior compartment is a prerequisite for its signaling activity in cells of the anterior compartment (*Crozatier et al., 2004*). To this end, comparable amounts of Hh and Hh variants (*Figure 2b*, *Supplementary file 1*) were expressed from one specific *attP 51C* landing site on the second chromosome (*Bateman et al., 2006*) using the Gal4/UAS system (*Brand and Perrimon, 1993*). In the posterior compartment, Hh was expressed under *en-Gal4* control, which is referred to as *en >Hh*, while in the anterior compartment, Hh was expressed in a stripe of cells under *ptc-Gal4* control, referred to as *ptc >Hh*. As previously shown (*Crozatier et al., 2004*; *Lee et al., 2001*; *Mullor et al., 1997*; *Strigini and Cohen, 1997*), *en >Hh* expanded the L3-L4 intervein area and, as a

concomitant effect, reduced the L2-L3 intervein space (*Figure 2c*). By contrast, *en*-regulated overexpression of non-palmitoylated, biologically inactive Hh$^{C85S}$ (*en >Hh$^{C85S}$*) resulted in L3-L4 veins being proximally apposed and the formation of ectopic anterior cross veins (*Figure 2d*) (*Crozatier et al., 2004*; *Lee et al., 2001*), suggesting that Hh$^{C85S}$ competes with bioactive wild-type Hh (*Lee et al., 2001*). This phenotype is consistent with the higher Hh concentrations required for activation of the target genes, *ptc* and *collier*, and L3-L4 development, than those required for the activation of *dpp*, which patterns the remainder of the wing (*Hooper, 2003*; *Méthot and Basler, 1999*; *Mohler et al., 2000*; *Strigini and Cohen, 1997*; *Vervoort et al., 1999*). Wing phenotypes were quantified by dividing the proximal L3-L4 areas by the L2-L3 areas (*Figure 2e,f*). This revealed significant Hh gain of function upon Hh overexpression in the posterior compartment or loss of function upon Hh$^{C85S}$ overexpression (*en >GFP* served as a normal control: L3-L4/L2-L3 = 0.074 ± 0.002; *en >Hh* = 0.108 ± 0.005 (+46%), p<0.0001; *en >HhC$^{C85S}$*=0.032 ± 0.002 (-57%), p<0.0001).

## Hh$^{C85S}$ cell-autonomously suppresses endogenous Hh function

To investigate the molecular basis of the dominant-negative Hh$^{C85S}$ activity in wing disc tissues, we spatially disconnected Hh$^{C85S}$ expression from endogenous Hh expression by using *ptc >Hh$^{C85S}$*. In the event that biologically inactive Hh$^{C85S}$ would impair the response to Hh in a non-cell autonomous manner, for example, by binding to and blocking the receptor Ptc, *ptc >Hh$^{C85S}$* wing phenotypes should be comparable, or even more severe than those observed in *en >Hh$^{C85S}$* wings. Alternatively, if unprocessed N-terminal Hh$^{C85S}$ peptides directly inhibit Ptc binding of associated endogenous Hh produced in the same compartment, we expected *ptc >Hh$^{C85S}$* wing phenotypes to be less severe than those observed in *en >Hh$^{C85S}$* wings. Indeed, Hh$^{C85S}$ expression under ptc control had little effect on wing development (*Figure 2g,h*: *ptc >GFP*: 0.084 ± 0.001, *ptc >Hh$^{C85S}$*: 0.078 ± 0.002 (-7%), p=0.0026), suggesting that Hh$^{C85S}$ cell-autonomously interferes with endogenous Hh, possibly by the random mixing of inactive Hh$^{C85S}$ and wild-type Hh at the cell surface (*Figure 2i*, left). In this mixed association, unprocessed Hh$^{C85S}$ N-terminal peptides block wild-type Hh-receptor-binding sites in trans. By contrast, ptc>Hh$^{C85S}$ expression in the anterior compartment prevents mixed cluster formation and therefore does not affect the controlled secretion and signaling of endogenous Hh (*Figure 2i*, right).

## Monomeric soluble HhN$^{C85S}$ does not inhibit wild-type Hh function

To independently confirm that Hh$^{C85S}$ dominant-negative function requires direct Hh/Hh$^{C85S}$ association with the same clusters, we expressed unlipidated monomeric HhN$^{C85S}$ (*Porter et al., 1996a*) in vitro and in vivo (*Figure 3a*). We observed that the expression of soluble HhN$^{C85S}$ under en-control did not affect endogenous Hh function in vivo (*Figure 3b*), as expected from its exclusion from lipidated Hh clusters at the cell surface. As shown in *Figure 3c*, relative L3-L4/L2-L3 ratios obtained from three independent HhN$^{C85S}$ lines (1-3) revealed that wing development was not significantly affected (*en >GFP*: 0.074 ± 0.002, *en >HhN$^{C85S}$* (1): 0.07 ± 0.001 (-5%) (p=0.0707), *en >HhN$^{C85S}$* (2): 0.074 ± 0.001 (±0%) (p=0.9419), *en >HhN$^{C85S}$* (3): 0.075 ± 0.002 (+1%) (p=0.6050), 20 wings were quantified in each line). Notably, HhN$^{C85S}$ expression under *ptc* control resulted in a small, yet significant gain-of-function phenotype (*Figure 3d,e*, *ptc >GFP*: 0.084 ± 0.001, *ptc >HhN$^{C85S}$* (1): 0.091 ± 0.002 (+8%) (p=0.0013), *ptc >HhN$^{C85S}$* (2): 0.089 ± 0.002 (+6%) (p=0.0348), *ptc >HhN$^{C85S}$* (3): 0.093 ± 0.001 (+11%) (p<0.0001), 20 wings were quantified in each line). This is consistent with the concept that Hh inactivation by adjacent unprocessed N-terminal peptides in trans is restricted to clustered, but not unclustered proteins (*Ohlig et al., 2011*; *Ohlig et al., 2012*). We conclude that the lack of Hh inhibition by monomeric HhN$^{C85S}$ (*Figure 3f*), even if expressed in the same cells, is consistent with required direct association of palmitoylated and non-palmitoylated morphogens for dominant-negative Hh$^{C85S}$ function.

## N-terminal Hh$^{C85S}$ truncation reverses its dominant negative activity over Hh

We next determined the molecular basis of the cell-autonomous inhibitory activities of Hh$^{C85S}$. In vitro, N-terminal peptides block Ptc-receptor-binding sites of adjacent Hh molecules in the cluster in trans (*Figure 4—figure supplement 1a–c*) (*Ohlig et al., 2011*). Thus, we predicted that N-terminal truncation of Hh$^{C85S}$ should restore Hh biofunction in mixed clusters. To test this idea and to mimic

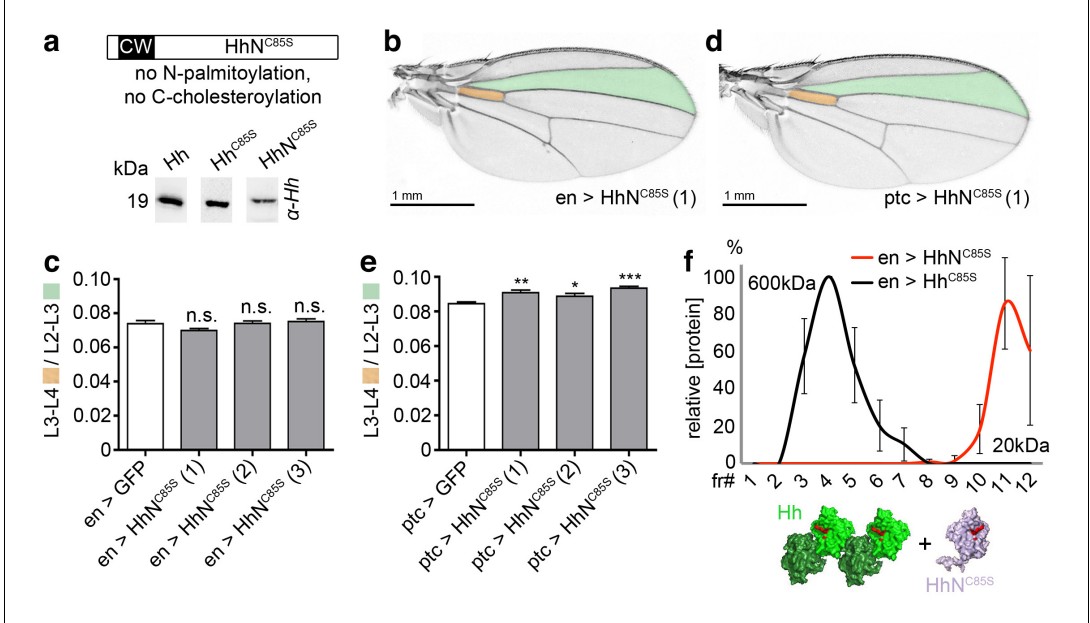

**Figure 3.** Unlipidated monomeric HhN[C85S] does not repress the formation of Hh-dependent wing structures. (**a**) Schematic of HhN[C85S] and compared expression of recombinant Hh, Hh[C85S] and HhN[C85S] in *Drosophila* S2 cells. Proteins were detected by α-Hh antibodies. (**b**) Unimpaired *Drosophila* wing formation upon *en*-controlled HhN[C85S] overexpression at 25°C. A representative result from one of four independently derived HhN[C85S] fly lines is shown. (**c**) Relative L3-L4/L2-L3 intervein ratios of three independent HhN[C85S] fly lines (1-3) are expressed as a quantitative readout for Hh patterning activity. ***$p \leq 0.001$, **$p \leq 0.01$, *$p \leq 0.05$, n.s. (not significant): $p > 0.05$, n = 20. (**d**) *Drosophila* wing formation upon *ptc*-controlled HhN[C85S] overexpression at 25°C. (**e**) Relative L3-L4/L2-L3 intervein ratios of three independent HhN[C85S] fly lines (1-3). ***$p \leq 0.001$, **$p \leq 0.01$, *$p \leq 0.05$, n.s. (not significant): $p > 0.05$, n = 20. (**f**) Gel filtration analysis of Hh[C85S] and HhN[C85S] expressed in *Drosophila* larvae under *en*-control. Multimeric Hh[C85S] eluted in fractions 3–8, corresponding to molecular weights of 70 kDa-600 kDa, as expected. By contrast, HhN[C85S] eluted in fractions 10–12 (corresponding to 19 kDa monomers). Elution profiles are expressed relative to the highest protein amounts detected, which were set to 100%. n = 3. Bottom: Proposed generation of endogenous Hh clusters (green) at the posterior cell surface that are devoid of monomeric soluble HhN[C85S].

DOI: https://doi.org/10.7554/eLife.33033.006

Hh processing observed in L3 *Drosophila* larvae (**Figure 4—figure supplement 1d,e**), we consecutively deleted N-terminal amino acids 86–91 (Hh[C85S;Δ86-91]) to 86–100 (Hh[C85S;Δ86-100]) (**Figure 4a**) and confirmed unimpaired protein expression (**Figure 4b**) and multimerization (**Figure 4—figure supplement 2**). All ten constructs were then inserted into the *attP-51C* landing site on the second chromosome to ensure comparable expression. At least three independent transgenic fly lines were derived from each construct and crossed with the *en-Gal4* driver line. We observed unchanged or moderately changed L3-L4/L2-L3 intervein ratios between *en* >Hh[C85S] and *en* >Hh[C85S;Δ86-91] to *en* >Hh[C85S;Δ86-97] adult wings (**Figure 4c–e,i** and **Figure 4—figure supplement 3a**). However, protein truncation beyond residue R97 gradually restored the biological activity of mixed clusters: *en* >Hh[C85S;Δ86-98] and *en* >Hh[C85S;Δ86-99] fly wings showed partially restored wing patterning (**Figure 4f,g,i**) and, strikingly, the posterior expression of Hh[C85S;Δ86-100] fully restored normal wing patterning (**Figure 4h,i** and **Figure 4—figure supplement 3a**). Cell-autonomous inhibitory activities of Hh[C85S] and restored wing patterning upon targeted coexpression of *en* >Hh[C85S;Δ86-100] were confirmed with the independent en-driver lines en(2)-Gal4 and hh-Gal4, both controlling transgene expression in the wing disc (**Figure 4—figure supplements 4 and 5**). These results are consistent with the assembly of Hh clusters by direct protein-protein contacts as a prerequisite for the inhibitory activity of unprocessed N-terminal peptides.

We also observed that wing phenotypes varied between and within fly lines (labeled 1–4 in **Figure 4—figure supplement 3b**). This variability can be explained by slightly different expression levels or by small stochastic changes in Hh/Hh[C85S] cluster composition with increasing relative amounts of Hh[C85S], resulting in stronger dominant negative phenotypes. Indeed, temperature-dependent Gal4-regulated transgene amounts (**Duffy, 2002**) affected dominant-negative wing phenotypes: At 29°C, increased amounts of Hh[C85S] relative to (fixed) endogenous Hh inhibited Hh function more

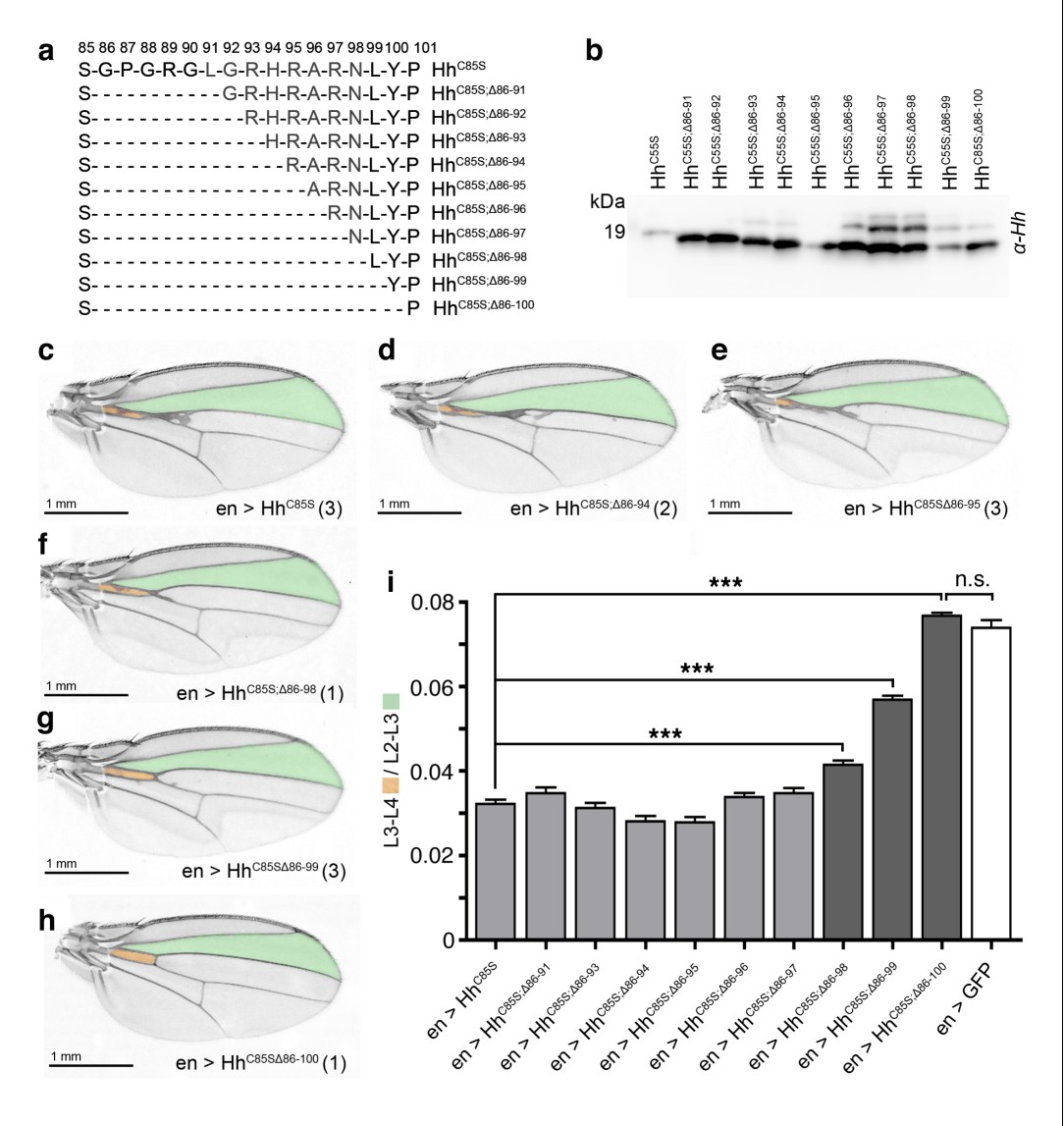

**Figure 4.** N-terminal truncation of palmitoylation-deficient Hh$^{C85S}$ rescues wing formation. (a) All truncated proteins also lacked the N-terminal cysteine, preventing Hh palmitoylation (*Hardy and Resh, 2012*). Residues #93–97: CW motif. (b) All proteins were expressed and secreted from S2 cells, as determined by immunoblotting. (c–h) *En*-regulated overexpression of Hh$^{C85S}$ and N-terminally truncated proteins (Hh$^{C85S;\Delta}$). Unaffected wing development despite *en*-regulated expression of unpalmitoylated Hh$^{C85S;\Delta86-100}$ (h). (i) Quantification of wings shown in c-h. *En*-regulated GFP and Hh$^{C85S}$ expressions served as positive and negative controls, respectively. Pooled analysis of three transgenic fly lines, each derived from an independent injection. *en >Hh$^{C85S}$*: 0.032 ± 0.001, *en >Hh$^{C85S;\Delta86-91}$*: 0.035 ± 0.001 (p=0.1375), *en >Hh$^{C85S;\Delta86-93}$*: 0.031 ± 0.001 (p=0.5458), *en >Hh$^{C85S;\Delta86-94}$*: 0.028 ± 0.001 (p=0.0134), *en >Hh$^{C85S;\Delta86-95}$*: 0.028 ± 0.001 (p=0.001), *en >Hh$^{C85S;\Delta86-96}$*: 0.034 ± 0.001 (p=0.25), *en >Hh$^{C85S;\Delta86-97}$*: 0.035 ± 0.001 (p=0.117), *en >Hh$^{C85S;\Delta86-98}$*: 0.041 ± 0.001 (p<0.0001), *en >Hh$^{C85S;\Delta86-99}$*: 0.057 ± 0.001 (p<0.0001); *en >Hh$^{C85S;\Delta86-100}$*: 0.076 ± 0.0007 (p=0.0001), en >GFP: 0.074 ± 0.002. ***p≤0.001, n.s. (not significant): p>0.05, n = 60 (n = 20 per line), all flies developed at 25°C.

DOI: https://doi.org/10.7554/eLife.33033.007

The following figure supplements are available for figure 4:

**Figure supplement 1.** Modeled linear *Drosophila* Hh clusters, using pdb 2IBG and pdb 3M1N as templates.

DOI: https://doi.org/10.7554/eLife.33033.008

**Figure supplement 2.** Unimpaired multimerization of N-terminally truncated Hh variants.

DOI: https://doi.org/10.7554/eLife.33033.009

*Figure 4 continued on next page*

*Figure 4 continued*

**Figure supplement 3.** Graded variable wing defects as a consequence of full-length and N-terminally truncated Hh$^{C85S}$ expression in the wing disc.

DOI: https://doi.org/10.7554/eLife.33033.010

**Figure supplement 4.** Confirmation that N-truncated Hh$^{C85S}$ and HhN$^{C85S}$ do not repress the formation of Hh-dependent wing structures, using the driver line hh-Gal4.

DOI: https://doi.org/10.7554/eLife.33033.011

**Figure supplement 5.** Confirmation that N-truncated Hh$^{C85S}$ does not repress the formation of Hh-dependent wing structures, using the driver line en(2)-Gal4.

DOI: https://doi.org/10.7554/eLife.33033.012

**Figure supplement 6.** Temperature-dependent dominant-negative Hh$^{C85S;\Delta86-93}$ function in the posterior *Drosophila* wing disc compartment.

DOI: https://doi.org/10.7554/eLife.33033.013

strongly, whereas reduced transgene expression at 18°C inhibited Hh function less strongly (***Figure 4—figure supplement 6***).

Taken together, we conclude that N-palmitate serves to ensure reliable membrane-association of inhibitory N-termini, making quantitative peptide processing a prerequisite for the solubilization of fully activated clusters (***Figure 5a***). As a consequence, Hh concentrations at any position in the gradient will strictly correlate with their biological activities (i.e. their Ptc-binding capacities). Impaired N-palmitoylation in this scenario reduces Hh bioactivity to variable degrees, depending on the relative number of unprocessed N-terminal peptides in soluble clusters (***Figure 5b***). This essential 'cleavage/activation control' function is confirmed by fully restored Hh biofunction upon targeted coexpression of *en* >Hh$^{C85S;\Delta86-100}$ (***Figure 5c***).

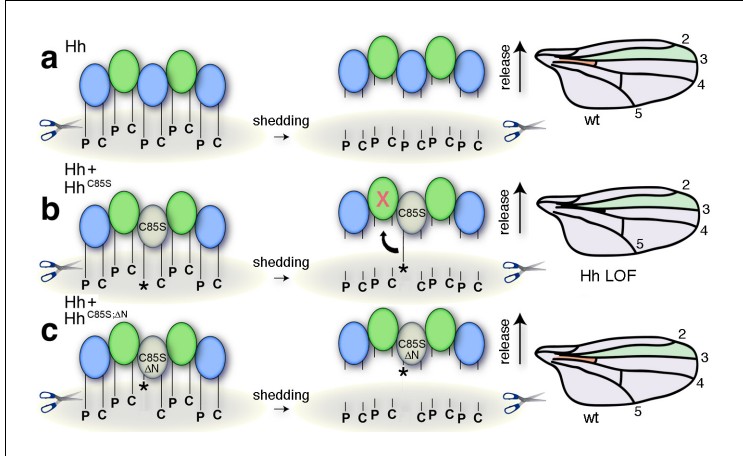

**Figure 5.** Simplified model for the conversion of membrane-bound Hh into soluble clusters. Surface-tethered wild-type Hh monomers (colored green and blue for clarity) form multimeric clusters with their extended N- and C-terminal lipidated peptides tethered to the cell membrane. (**a**) Membrane-proximal proteolytic processing (scissors) removes lipidated membrane anchors and releases Hh clusters from posterior cells. As a consequence, protein concentrations at any position in the responsive (anterior) field correlate with their biological activities (their Ptc-receptor-binding capacities). Because partially processed clusters are not released, the role of both lipids at this point is to control quantitative Hh processing and bioactivation. (**b**) Unpalmitoylated Hh$^{C85S}$ only requires processing of its cholesterylated C-terminus for release: As a consequence, a fraction of wild-type Hh in mixed clusters has its Ptc-receptor-binding sites and bioactivity blocked (indicated by the X) by unprocessed adjacent Hh$^{C85S}$ N-termini (asterisk). Signaling at any position in the field is thus reduced, leading to dominant negative wing phenotypes (right). LOF: loss of function. (**c**) Artificial truncation of unpalmitoylated Hh$^{C85S}$ N-termini restores wild-type Hh function. wt: wild type.

DOI: https://doi.org/10.7554/eLife.33033.014

## N-palmitate controls Hh release from the cell surface in vitro

In our model, N-palmitate tethers incompletely processed Hh clusters to the cell membrane to prevent their release. To test this hypothesis, we utilized a cell culture model employing Bosc23 cells. To achieve quantitative Hh N-palmitoylation in vitro, we used bicistronic mRNA constructs to couple Shh (the vertebrate Hh ortholog) and Hh acyltransferase expression in the same cells. We then compared the release of fully lipidated Shh, non-palmitoylated Shh$^{C25S}$, and variants carrying the extended C-terminal membrane anchor N$^{190}$SVAAKSG-*YPYDVPDYA*-G$^{198}$ (G$^{198}$ represents the cholesterol-modified glycine; italicized underlined letters represent the tag, *Figure 6*) (*Jakobs et al., 2014*). Proteins were detected by polyclonal α-Shh antibodies and monoclonal α-HA antibodies on the same (stripped) blots. Grayscale blots were inverted, colored (green: α-Shh signal, blue: α-HA signal) and merged to identify proteins bound by both antibodies (yielding bright blue/cyan signals) and proteins bound by only α-Shh antibodies (green signals).

As shown in *Figure 6a*, dual-lipidated Shh and Shh$^{HA}$ yielded strong cellular signals but were absent from media, indicating impaired release. By contrast, non-palmitoylated Shh$^{C25S;HA}$ was effectively converted into a C-terminally truncated soluble morphogen, as indicated by an electrophoretic size shift and lack of α-HA antibody reactivity (compare the cellular (c) material in each lane 1 with corresponding media in each lane 3). Three independent quantifications of dual-lipidated Shh, cholesterylated Shh$^{C25S}$, and non-lipidated ShhN$^{C25S}$ in cells and media (*Figure 6b*) confirmed that N-palmitoylation controls protein solubilization in vitro (1 hr release: Shh$^{C25A}$ 94 ± 3 arbitrary units (a.u.), ShhN$^{C25A}$ 265 ± 10 a.u., p<0.0001, n = 3; 4 hr release: Shh 18 ± 2 a.u., Shh$^{C25A}$ 238 ± 6 a.u., p>0.0001, n = 3, ShhN$^{C25A}$ 500 ± 58 a.u., p>0.001, n = 2; values express ratios between solubilized/cell-associated proteins). Accordingly, coexpression of dual-lipidated Shh and Shh$^{C25A}$ in the same Bosc23 cells resulted in mixed clusters and thereby a four-fold reduction in Shh$^{C25A}$ release (Shh+Shh$^{C25A}$: 25.7 ± 5%, Shh$^{C25A}$ alone was set to 100%, p<0.0001, n = 7) (*Figure 6c*). Importantly, we further observed that dual-lipidated, N-terminally HA-tagged $^{HA}$Shh was not released (*Figure 6d*). In this construct, the HA tag was inserted at the position of the membrane-proximal CW motif, shifting this previously identified sheddase target site (*Ohlig et al., 2012*) distally while not affecting its HS-binding capacity.

To test whether the same modification would also impair release of fly Hh, we inserted an HA tag between corresponding Hh amino acids L91 and G92, resulting in the N-terminal $^{HA}$Hh sequence C$^{85}$GPGRGL$^{91}$-*YPYDVPDYAG*$^{92}$-**RHRAR**N (bold letters represent the CW motif that is shifted nine amino acids away from the preferred membrane proximal site of sheddase activity). We also used Hh, non-palmitoylated Hh$^{C85S}$ and Hh$^{C85S;Δ86-100}$ as controls. $^{HA}$Hh was expressed in S2 cells, its unimpaired multimerization confirmed (*Figure 6—figure supplement 1*), and cellular and soluble proteins compared by SDS-PAGE and immunoblotting (*Figure 6e*). We observed that all proteins were produced in S2 cells, as indicated by strong α-Hh antibody binding to all cellular forms. In contrast, only low levels of $^{HA}$Hh that retained the tag were solubilized, suggesting that N-terminal processing was impaired in S2 cells. From these experiments, we conclude that N-palmitoylation controls Hh release from the cell surface and restricts possible modes of Hh solubilization to shedding (*Figure 6f,g*).

## Unimpaired proteolytic processing of N-palmitoylated Hh termini is essential for direct but not indirect in vivo Hh signaling

We next generated transgenic flies expressing $^{HA}$Hh (*Figure 7a*) in the posterior compartment. The HA-tagged protein, due to its invariable association with the membrane (*Figure 6e*) and direct association with endogenous Hh in mixed clusters, was expected to impair endogenous Hh release and to lead to severe dominant-negative mis-patterning phenotypes. Indeed, $^{HA}$Hh expression in the posterior compartment at 25°C largely arrested fly development at the pupal and pharate stages, leading to defective head development characteristic of Hh loss of function (*Torroja et al., 2004*). Of 230 pharates counted, only three imagos hatched with smaller wings lacking anterior structures (*Figure 7b*), again characteristic of Hh loss of function (*Bejarano et al., 2012*). Reduced transgene expression at 18°C largely reversed pharate lethality: 77% of en> $^{HA}$Hh pharates hatched (293 flies from 393 pupae) but wing development was still impaired with all analyzed wings lacking all or most of the L3-L4 intervein area (*Figure 7c*). This phenotype resembles wing phenotypes of flies expressing non-diffusible HhCD2 (*Strigini and Cohen, 1997*) or with impaired activity of Hh signaling

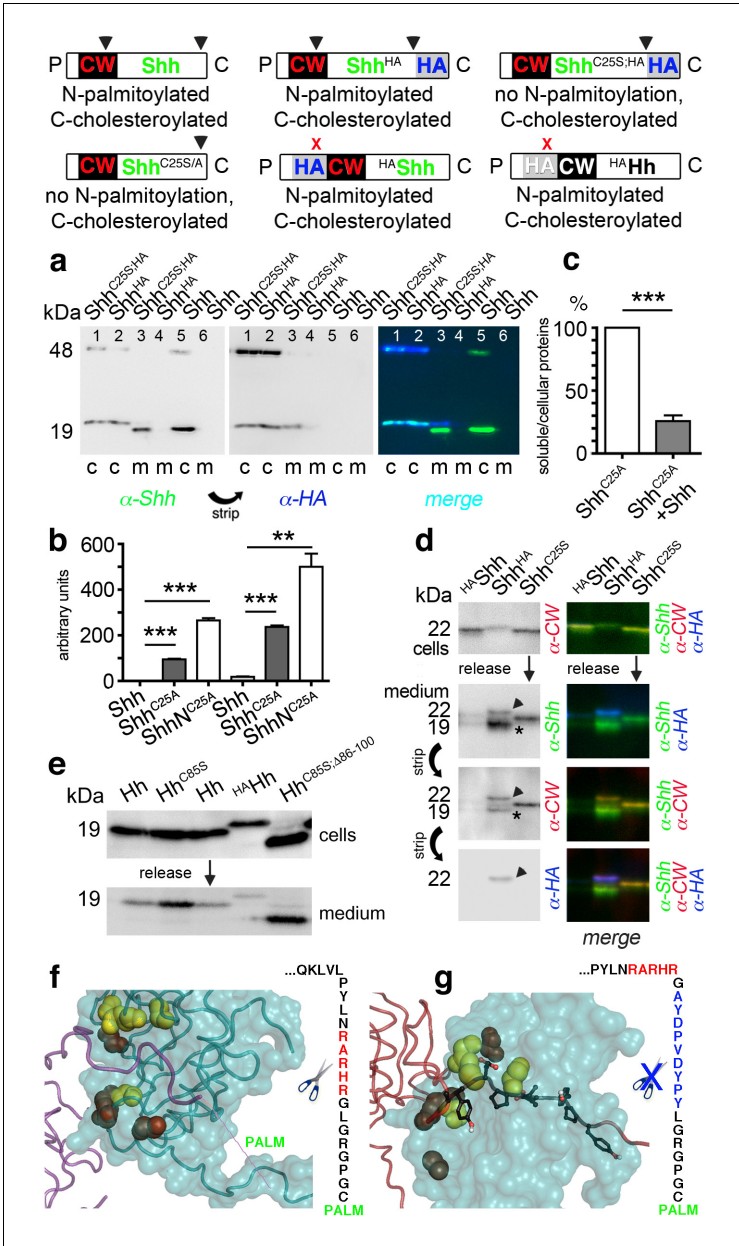

**Figure 6.** N-palmitoylation controls Hh morphogen release in vitro. Top: Schematics of transgenes. Arrowheads indicate cleavage sites, the x denotes blocked cleavage. (**a**) Palmitoylated vertebrate Hh orthologs Shh and Shh[HA] and non-palmitoylated Shh[C25A;HA] were expressed in Bosc23 cells, and the proteins in the cellular (c) and corresponding soluble fractions (m) were analyzed by immunoblotting. To better demonstrate protein processing during release, we inverted grayscale blots and colored them (right: green: α-Shh, blue: α-HA). Green signals label untagged or processed Shh, and cyan signals label unprocessed HA-tagged proteins. Higher electrophoretic mobility confirms terminal processing during release. Tagged and untagged palmitoylated proteins are not efficiently released. (**b**) Compared dual-lipidated, monolipidated and non-lipidated Shh release after 1 hr (left) and 4 hr (right). ***p≤0.0001, **p≤0.001, n = 3. (**c**) Release of Shh[C25A] is downregulated 4-fold upon dual-lipidated Shh coexpression (27.7 ± 4.6% if compared to Shh[C25A] release alone, which was set to 100%, n = 7, ***p≤0.0001). (**d**) Impaired release of N-terminally HA-tagged [HA]Shh. α-CW antibodies detect the N-terminal CW motif (KRRHPKK). Bright cellular signals in merged blots denote unprocessed proteins (arrowhead), orange signals denote C-processed/N-unprocessed proteins, and green signals confirm the removal of N- and C-terminal peptides (asterisk). Note Shh[HA] processing at the CW site during release. By contrast, N-terminal processing of Shh[C25S] is impaired. (**e**) Immunoblot analysis of recombinant Hh proteins released into media of transfected *Drosophila* S2 cells (left). Top row: S2 cells express palmitoylated and non-palmitoylated proteins to comparable levels. Bottom

*Figure 6 continued on next page*

*Figure 6 continued*

row: S2 cells released high levels of unpalmitoylated Hh[C85S] and Hh[C85S;Δ86-100], and lower levels of palmitoylated Hh, into the media. Only very low levels of [HA]Hh were released in unprocessed form (top band). (**f**) Intermolecular interactions of *Drosophila* Hh N-terminal peptides. Right: schematic of the palmitoylated N-terminal 'stem' peptide, including basic CW residues (red) serving as the predicted membrane-proximal cleavage site. (**g**) Modeled insertion of the HA tag upstream of the N-terminal CW motif (located between Hh residues L91 and G92) of *Drosophila* Hh. This moves the CW motif nine amino acids more distal to the membrane and replaces its previous position with the HA tag (blue, right). Modeled N-terminal palmitate is shown as a zigzag line (pointing to the right). Yellow spheres denote *Drosophila* Hh residues corresponding to Shh residues that interact with Ptc (*Drosophila* Hh H193, H194, H200, H240) (**Bosanac et al., 2009**). Red spheres denote residues corresponding to Shh amino acids bound by the Shh inhibitory antibody 5E1 (K105, R147, R213, R238, R239 in *Drosophila* Hh) (**Maun et al., 2010**).

DOI: https://doi.org/10.7554/eLife.33033.015

The following figure supplement is available for figure 6:

**Figure supplement 1.** Unimpaired multimerization of HA-tagged Hh.

DOI: https://doi.org/10.7554/eLife.33033.016

components such as Fused or Collier (Col) (**Ascano and Robbins, 2004**; **Vervoort et al., 1999**), while the distal 'widening' of L3 (**Figure 7c**) is consistent with impaired Hh repression of Iroquois-regulated L3 formation (**Crozatier et al., 2004**). Notably, the observation that [HA]Hh expression under *ptc*-control showed only minor effects (**Figure 7h,i**) confirms cell-autonomous Hh repression by direct [HA]Hh contacts in mixed clusters, and suggests that palmitoylated [HA]Hh peptides restrain these mixed clusters at the cell membrane. We therefore expected that additional C > S

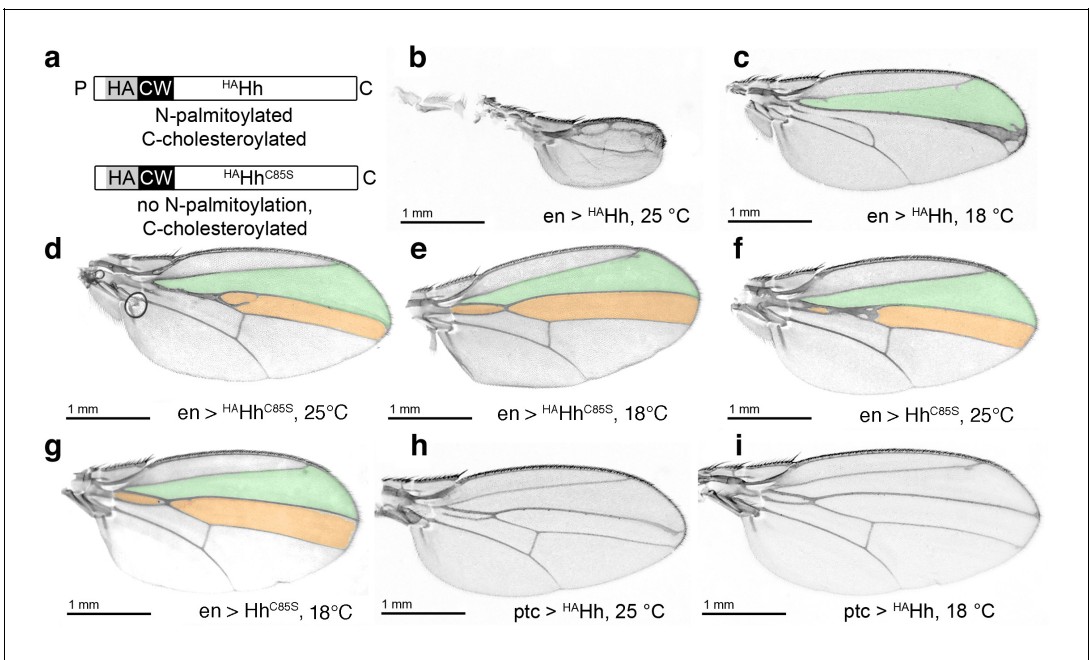

**Figure 7.** HA tag insertion into the putative N-terminal processing site strongly represses wild-type Hh in vivo. (**a**) Schematic of HA-tagged Hh constructs. P: palmitate, C: cholesterol, CW: CW motif, HA: HA tag. (**b**) At 25°C, most flies die at the late larval/early pupal stage. Wings of the few surviving *en* >[HA]Hh flies show severe dominant-negative Hh loss of function. This phenotype was observed in four fly lines derived from two [HA]Hh integration events each into *VK37, 51C* and *attP* integration sites. (**c**) At 18°C, more flies develop, and L3 and L4 appose into a large central vein. (**d,e**) Additional deletion of the palmitate acceptor cysteine ([HA]Hh[C85S]) largely reverses Hh loss of function. At 25°C, wings show proximally apposed L3-L4 veins, and at 18°C, the anterior crossvein is reduced, as previously observed for non-palmitoylated Hh[C85S]. (**f,g**) Representative *en* >Hh[C85S] wing phenotypes are shown. (**h,i**) By contrast, *ptc*-controlled [HA]Hh expression in the anterior (Hh-receiving) wing disc compartment at 18°C and 25°C mildly affected wing formation.

DOI: https://doi.org/10.7554/eLife.33033.017

mutagenesis, by removing the membrane anchor (*Figure 7a*), would revert the observed severe mis-patterning phenotypes due to impaired cluster release into milder forms caused by partially impaired Hh binding to Ptc, as described earlier. Indeed, additional mutagenesis of the palmitate acceptor cysteine in *en >*HHhC85S flies fully reversed pharate lethality at 25°C and led to wing phenotypes comparable to those of *en >HhC85S* flies (compare *Figure 7d* with 7 f and *Figure 7e* with 7 g).

## Repressed target gene transcription by N-terminally unprocessed palmitoylated and non-palmitoylated Hh

The Hh gradient emanating from the posterior compartment activates the Hh target genes *engrailed* (*en*), *collier* (*col*), *patched* (*ptc*) and *decapentaplegic* (*dpp*) in stripes of anterior cells adjacent to the a/p border. Via Hh-responsive accumulation and nuclear access of Ci[155], *en* and *col* are induced in a 5- to 7 cell wide anterior stripe, *ptc* in a 10 cell wide stripe and *dpp* in a 12–15 cell wide stripe, and the presence and width of these stripes of target gene expression is differentially sensitive to Hh dose (*Chen and Struhl, 1996*; *Strigini and Cohen, 1997*). Far from the Hh source, Ci[155] is depleted to form the repressor Ci[R], and Hh target genes are repressed. Cells receiving minimal amounts of Hh activate *dpp* transcription, cells receiving an intermediate amount of Hh activate the expression of *col* and *ptc* in addition to that of *dpp*, and Hh-dependent anterior *en* transcription (but not posterior, Hh-independent *en* transcription) is located closest to the a/p border (*Figure 8a,b*). *Col* in the high and intermediate zones down-regulates Dpp responses: This results in the future L3-L4 intervein (*Mohler et al., 2000*; *Vervoort et al., 1999*). We used this system to investigate the impact of our mutant forms of Hh on the expression of *en*, *ptc* and *dpp*. We confirmed that posterior Hh overexpression expanded *dpp-LacZ* expression anteriorly (*Figure 8c*) and, consistent with established *en >Hh* expansion of the L3-L4 intervein area (*Figure 2c*), we confirmed that posterior Hh overexpression expanded *ptc-LacZ* expression in the presumptive L3-L4 region (*Figure 8d*). En-controlled expression of the HA-tagged protein at 18°C, in contrast, did not much affect *dpp-LacZ* expression in the anterior compartment (*Figure 8e*), but abolished all *ptc-LacZ* reporter expression and restricted *en*-expression posteriorly (*Figure 8f*). This confirms that the complete loss of L3-L4 intervein tissue in adult *en >*HHhh wings is caused by insufficient Hh levels at the a/p border, and supports the idea that coexpressed HHhh impaired Hh release from the posterior compartment of the wing disc. We note that the observed expansion of *dpp* expression can be best explained by abrogated Ptc-mediated Hh internalization that normally restricts the Hh gradient (*Chen and Struhl, 1996*). Consistent with our concept of N-palmitate serving as a membrane anchor to prevent unregulated Hh release, HHhhC85S coexpression restored *ptc-LacZ* and *dpp-LacZ* expression (*Figure 8g,h*) to levels comparable to those detected in HhC85S expressing wing discs (*Figure 8i,j*), and additional deletion of the unpalmitoylated N-terminal peptide reverted the expanded area of *dpp-LacZ* expression to wild-type range (*Figure 8k,l*). Together, these experiments confirm that N-terminal Hh processing converts the insoluble Hh cluster into truncated, bioactive morphogen, and that the palmitate anchor controls completion of this process.

## Dominant-negative HHhhC85S activity on Ptc binding, but not dominant-negative HHhh activity, is compensated by increased Hh coexpression

To confirm that impaired processing of palmitoylated Hh variants prevents their solubilization, while impaired processing of unpalmitoylated Hh N-termini affects Ptc receptor binding of soluble clusters (merely reducing their bioactivity), we macroscopically analyzed wings of single and compound transgenic fly lines expressing Hh from the *attP 51C* landing site on chromosome 2 and HHhh or HHhhC85S from one specific *attP2* landing site on chromosome 3. As shown earlier, if expressed under *en-Gal4* control, Hh and HHhhC85S strongly affected wing development: En-controlled HHhhC85S reduced the formation of L3-L4 intervein tissue (*Figure 9a*), and *en >Hh* expanded the L3-L4 intervein area (*Figure 9b*). Targeted coexpression of both proteins fully reverted dominant-negative HHhhC85S function in 80% of wings and expanded this area in the remaining 20% of wings (56 wings were analyzed, *Figure 9c,d*). This indicates that increased Hh amounts 'titer out' dominant-negative HHhhC85S function. In contrast, *en >Hh* did not significantly correct dominant-negative HHhh mis-patterning phenotypes. As previously shown, HHhh expression in the posterior compartment arrested fly development at pupal and pharate stages. All wings of about 4% imagos that

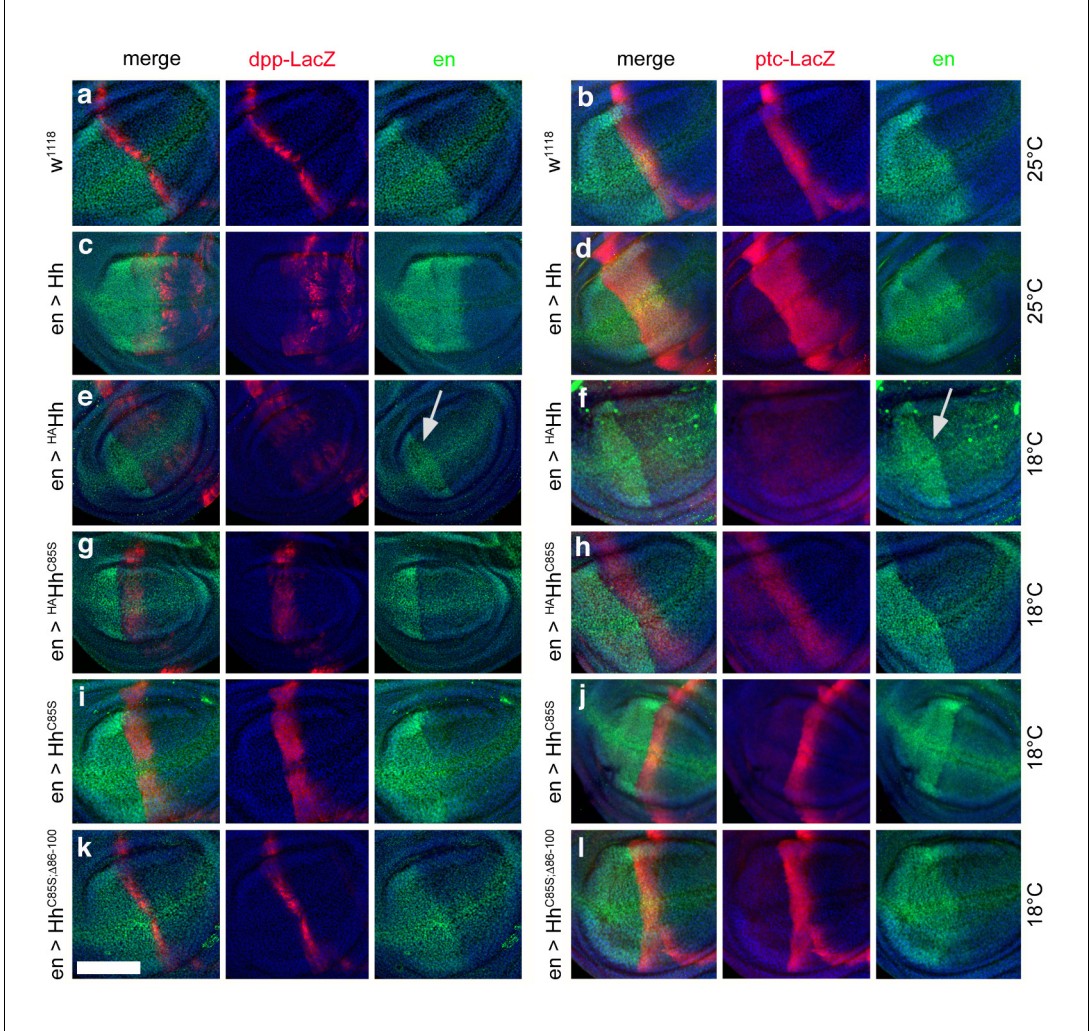

**Figure 8.** Effect of different Hh variants on *en*, *dpp-LacZ* and *ptc-LacZ* expression in the wing disc (**a,b**) *Dpp-LacZ* (**a**) and *ptc-LacZ* (**b**) reporter gene expression at the a/p border in wild-type third-instar discs. Nuclear β-galactosidase is immunofluorescently labeled (red). Overexpression of CD8-GFP under *en*-control labels the posterior compartment. Fly larvae developed at 25°C. The left image is a merge. (**c,d**) Hh overexpression expands *dpp-LacZ* and *ptc-LacZ* expression anteriorly. 4D9 α-engrailed/invected (inv) antibodies label the posterior compartment in this and the following panels (green). Fly larvae developed at 25°C. (**e,f**) *En*-controlled $^{HA}$Hh overexpression reduced *dpp-LacZ* expression in the anterior wing disc. *Ptc-LacZ* expression was always completely abolished, and *en/inv* expression was restricted to most posterior wing disc regions (arrow). Fly larvae for this and all subsequent analyses developed at 18°C because wing disc growth arrested at 25°C, preventing further analysis. (**g,h**) Additional deletion of the palmitate membrane anchor increased *dpp-LacZ* expression and also restored a stripe of weak yet expanded *ptc-LacZ* expression. The expansion of *ptc-LacZ* and *dpp-LacZ* domains beyond wild-type levels may be linked to reduced *inv/en* expression in anterior target cells (note the unchanged posterior restriction of *inv/en*-expression). (**i,j**) *En*-controlled Hh$^{C85S}$ expression leads to comparable *dpp-LacZ* expression. *Ptc-LacZ* reporter expression was reverted into more intense and restricted staining, indicating an additional inhibitory effect of the HA-tag. (**k,l**) Restored wild-type pattern of *dpp-LacZ* expression and *ptc-LacZ* expression as a consequence of *en*-controlled Hh$^{C85S;\Delta86-100}$ expression shows that expanded *dpp-LacZ* expression and reduced *ptc-LacZ* expression in *en >Hh$^{C85S}$* and *en >$^{HA}$Hh$^{C85S}$* discs were caused by the unprocessed N-terminal peptide. Wing discs are oriented such that anterior is right and dorsal is up; all magnification, camera and processor settings were kept identical. Scale bar: 100 μm.
DOI: https://doi.org/10.7554/eLife.33033.018

hatched (n = 12/320) lacked most anterior structures (***Figure 9d,e***). Hh coexpression partially reversed pharate lethality (22% imagos hatched, n = 41/182), but all analyzed wings still lacked the complete L3-L4 intervein area (***Figure 9d,f***). The most likely explanation for this is that increased relative Hh amounts 'dilute' the average number of permanent membrane anchors of any given mixed cluster, with only a limited compensatory effect on Hh release and activity due to the remaining tethers.

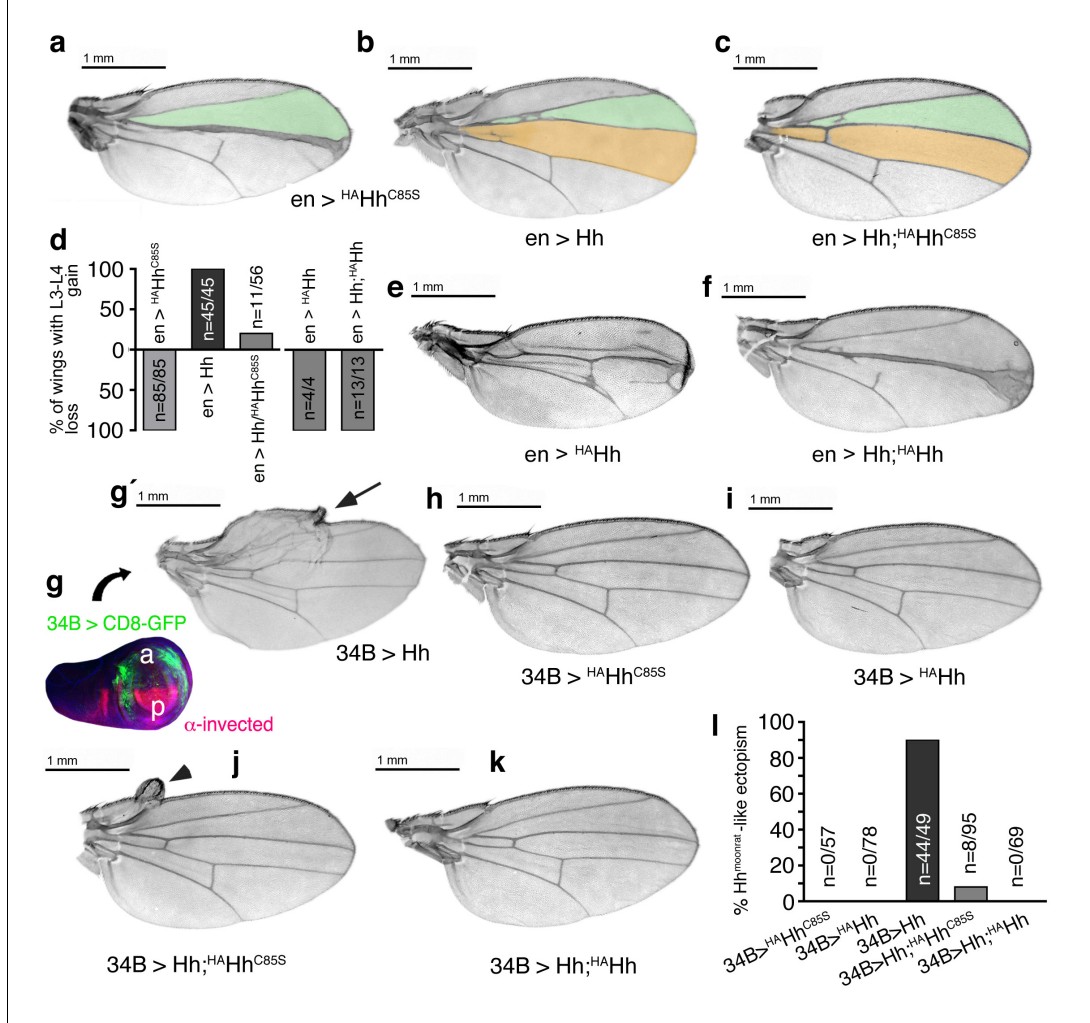

**Figure 9.** Increased Hh amounts compensate for impaired Ptc binding by unprocessed N-termini, but not for impaired Hh release in vivo. (a) If expressed from chromosome 3 at 25°C, en>$^{HA}$Hh$^{C85S}$ wings show proximally apposed L3-L4 veins. (b) En >$Hh$ wings show enlargement of L3-L4 intervein area. (c) En >$Hh$;$^{HA}$Hh$^{C85S}$ coexpression reversed en >$^{HAH}h$C$^{C85S}$ loss of function at 25°C and about 20% of wings showed Hh gain-of-function. Wing phenotypes are shown and quantifications shown in (d). (e) If expressed from chromosome 3 at 18°C, only 4% of en >$^{HA}$Hh flies hatch and show severe dominant-negative Hh loss-of-function phenotypes. (f) Upon Hh coexpression, at 18°C, 22% flies develop with their L3 and L4 always fused into one central vein. (g,g′) 34B-Gal4 expresses UAS-transgenes at the border of the anterior wing disc (green) that does not overlap with the posterior hh-producing disc compartment (red). 34B > Hh expression at 25°C led to clear anterior overgrowth in 90% of wings. (h,i) 34B > $^{HAH}h$C$^{C85S}$ or $^{HA}$Hh expression did not impair wing development. This again confirmed biological inactivity of both Hh variants. (j) 34B-controlled Hh;$^{HA}$Hh$^{C85S}$ coexpression partially reversed Hh gain-of-function, reducing ectopic overgrowth to a small fraction of wings (8%). (k) 34B > Hh;$^{HA}$Hh coexpression completely reversed Hh gain-of-function. This confirms cell-autonomous Hh repression by direct $^{HA}$Hh contacts in mixed clusters, as quantified in (l).
DOI: https://doi.org/10.7554/eLife.33033.019

We confirmed cell-autonomous Hh repression by using another Gal4-line (34B-Gal4) that drives transgene expression in cells that form the most proximal parts of the wing where *hh* is normally not expressed (***Figure 9g***)(***Brand and Perrimon, 1993***). *34B-Gal4*-controlled Hh expression in these cells results in phenotypes resembling a natural *hh* gain-of-function allele, *hh*$^{Moonrat}$ (***Figure 9g′***, arrow) (***Tabata and Kornberg, 1994***). Phenotypes resulting from ectopic *hh*$^{Moonrat}$ expression are usually mild, varying between overgrowth of anterior wing tissue to slight disorganization of the wing margin and the addition of extra vein material. We observed that *34B > $^{HA}$Hh$^{C85S}$* and *34B > $^{HA}$Hh* expression did not affect wing development, confirming spatial disconnection of 34B-directed transgene expression from posterior endogenous Hh production and biological inactivity of both proteins (***Figure 9h,i***). In compound *34B > Hh;$^{HA}$Hh$^{C85S}$* wings, the activity of mixed clusters was reduced

(*Figure 9j*, arrowhead), while it was completely abolished in *34B > Hh;$^{HA}$Hh* wings (*Figure 9k,l*). This is expected from impaired Ptc-binding of Hh;$^{HA}$Hh$^{C85S}$ clusters in the former situation versus blocked release of Hh;$^{HA}$Hh clusters in the latter situation.

## Reversed target gene expression recapitulates restored wing development in flies coexpressing Hh and $^{HA}$Hh$^{C85S}$

Finally, we investigated the expression of the Hh target gene ptc in flies expressing Hh and $^{HA}$Hh$^{C85S}$ alone and in combination. As shown earlier, posterior Hh overexpression expanded *ptc-LacZ* expression (compare *Figure 10a* with *Figure 10b*), and *en*-controlled expression of the HA-tagged non-palmitoylated protein strongly reduced *ptc-LacZ* reporter expression (*Figure 10c*). Consistent with the restored formation of L3-L4 intervein tissue in adult *en >Hh;$^{HA}$Hh$^{C85S}$* wings (*Figure 9c*), and occasionally gain-of-function in these wings, *ptc-LacZ* expression in the anterior compartment of the wing disc was expanded (*Figure 10d*). This shows that coexpressed Hh fully restored dominant-negative $^{HA}$Hh$^{C85S}$ function by expanding *ptc-LacZ* target gene expression in the presumptive L3-L4 region in the anterior compartment and demonstrates that receiving cells respond to the morphogen.

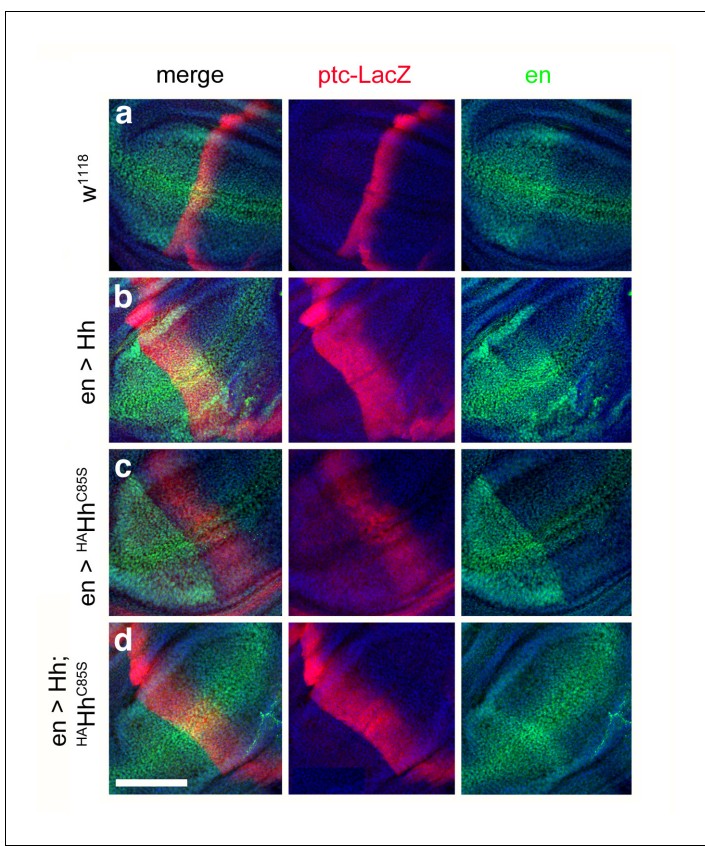

**Figure 10.** Combined en >Hh;$^{HA}$Hh$^{C85S}$ expression restores ptc-LacZ expression in the wing disc. (a) *Ptc-LacZ* reporter gene expression at the a/p border in wild-type third-instar discs. Nuclear β-galactosidase is immunofluorescently labeled (red). 4D9 α-engrailed/invected (inv) antibodies label the posterior compartment in this and the following panels (green). The left image is a merge. Fly larvae developed at 25°C. (b) En-controlled Hh overexpression increased and expanded *ptc-LacZ* expression anteriorly. (c) *En*-controlled $^{HA}$Hh$^{C85S}$ overexpression reduced *ptc-LacZ* expression, as previously shown. (d) *En*-controlled Hh;$^{HA}$Hh$^{C85S}$ coexpression generated a stripe of intense expanded *ptc-LacZ* staining, consistent with the gain-of-function phenotype observed in wings. Wing discs are oriented such that anterior is right and dorsal is up; all magnification, camera and processor settings were kept identical. Scale bar: 100 µm.

DOI: https://doi.org/10.7554/eLife.33033.020

Together, these experiments confirm a functional link between Hh lipidation, formation of linear cell surface clusters and proteolytic processing of lipidated N-terminal peptides in vivo. Processing serves to convert the lipidated morphogen cluster at the cell surface into the active form (*Figure 11*). Therefore, the N-palmitate membrane anchor and membrane-proximal CW-residues are functionally linked since the palmitoylation ensures quantitative CW-cleavage as a prerequisite for full Hh activation in vivo.

## Discussion

It is well established that cell-surface HS chains assist in Hh multimerization as a prerequisite for the generation of light microscopically visible aggregates at the cell surface (*Ortmann et al., 2015*; *Vyas et al., 2008*). Here, we provide ultrastructural data showing that a significant fraction of Hh assembles into extended linear arrays, consistent with the long unbranched HS-chain structure that scaffolds Hh clusters (*Vyas et al., 2008*), observed crystal lattice interactions of the vertebrate Shh ortholog (*Pepinsky et al., 1998*) and functional in vitro data (*Ohlig et al., 2011*; *Ohlig et al., 2012*). By exploiting the *Drosophila* wing development model which is dependent on differential Hh signaling for the formation of distinct wing structures, we further show that N-terminal peptides can block Ptc-receptor-binding of Hh clusters in vivo. Consistent with this, expression of N-truncated Hh mutants in *Drosophila* revealed that inhibitory peptide removal unmasks Ptc-binding sites and mediates direct, high threshold tissue patterning (*Strigini and Cohen, 1997*). Yet, contrary to previous observations on N-truncated Shh (*Ohlig et al., 2011*), we note that all artificially truncated Hh[C85S;Δ] variants were functionally inert. We explain this inactivity by Hh[C85S;Δ] misfolding due to possible intramolecular chaperone function of the 84 amino acid N-terminal Hh pre-peptide (*Eder and Fersht, 1995*) or unproductive Ptc binding of artificially truncated proteins as described for monomeric ShhN (*Williams et al., 1999*). We currently investigate these possibilities by insertion of a tobacco etch virus (TEV) protease recognition site into the putative Hh target site to allow for sequence-specific Hh[C85S] cleavage and activation after controlled TEV protease expression in the fly (*Harder et al., 2008*).

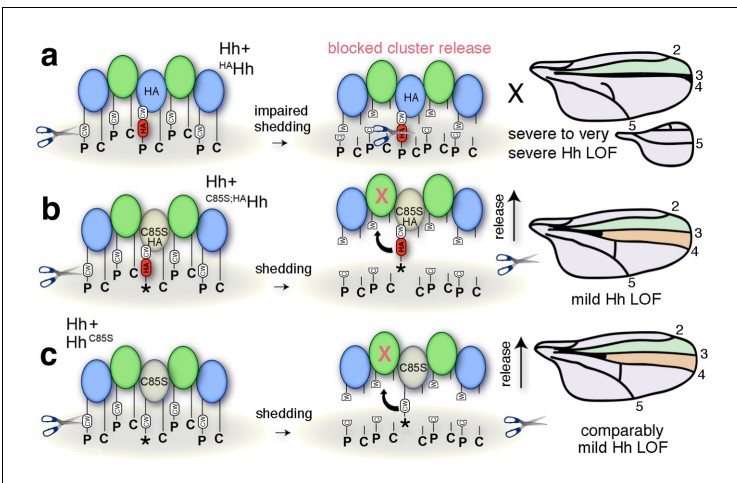

**Figure 11.** Simplified model for the impaired solubilization of membrane-bound Hh clusters containing [HA]Hh. (a) HA insertion in the predicted CW sheddase cleavage site strongly impedes wild-type Hh function. We explain this as completely blocked cell-surface release of all Hh morphogens in mixed clusters also containing unprocessed [HA]Hh. The X indicates blocked proteolytic processing. LOF: loss of function. (b) Additional removal of the N-terminal lipid anchor converts severely impaired wing development (and en >[HA]Hh larval lethality at 25°C) into milder phenotypes characteristic for unprocessed yet soluble clusters containing Hh[C85S] (c). This confirms that the N-terminal lipid anchor acts to control quantitative Hh processing and bioactivation during release and that dominant-negative Hh[C85S] function in wing development is not directly caused by the lack of N-palmitate.
DOI: https://doi.org/10.7554/eLife.33033.021

We previously showed that proteolytic conversion targets the N-terminal CW-site in vitro (*Dierker et al., 2009*; *Ohlig et al., 2012*). We show here that insertion of HA peptides, which displaces this cleavage site distally without affecting Hh N-palmitoylation (*Hardy and Resh, 2012*) and HS-dependent multimerization, is sufficient to impair endogenous and transgenic Hh high threshold biofunction in vivo, apparently without affecting low threshold Dpp-mediated Hh activity. Rescue of Hh biofunction by the additional mutation of the cysteine acceptor shows that N-palmitate anchors the unprocessed peptide to the cell membrane to safeguard Hh release. These findings are consistent with enhanced *Drosophila* Hh release upon RNAi-mediated knockdown of Hh acyltransferase activity (*Chamoun et al., 2001*) and increased Shh tethering to cell membranes by palmitate (*Konitsiotis et al., 2014*; *Levental et al., 2010*). Importantly, while S- and O-linked palmitate moieties are susceptible to enzymatic deacylation by palmitoyl-protein thioesterases (*Kakugawa et al., 2015*), amide-linked Hh palmitate is thioesterase resistant. This suggests that Hh relay from posterior subcellular structures – at least at some point – requires proteolytic processing of sheddase-accessible, membrane-proximal terminal target peptides. Support for this idea comes from the published replacement of the C-terminal Hh target peptide with transmembrane-CD2 (*Strigini and Cohen, 1997*). Resulting Hh-CD2 fusion proteins remain permanently membrane associated and generate wings with one single central vein in the region normally occupied by veins L3 and L4, while leaving Dpp-mediated anterior and posterior wing patterning intact. We note that this phenotype is strikingly similar to the *en >*$^{HA}$*Hh* phenotype described here. Moreover, required Hh transfer between protruding cell-cell contact structures emanating from the Hh-sending and Hh-receiving compartments, called cytonemes, was recently indicated by impaired Ptc signaling and internalization at contact sites with Hh-CD2 (*González-Méndez et al., 2017*). While the exact mechanism by which Hh is liberated from the posterior cytoneme membrane was not addressed, proteolytic Hh relay and reception at cytoneme contact sites was suggested by the authors, and is supported by the results shown in our work (*Figure 1e*).

In addition to cytoneme contact sites, other subcellular structures of P-compartment cells release Hh from the membrane (*Figure 12*). It has been suggested that the Hh gradient in *Drosophila* wing imaginal discs consists of apical and basolateral secreted pools formed as a consequence of initial apical Hh secretion, subsequent reinternalization, and apical (*D'Angelo et al., 2015*) or basolateral (*Callejo et al., 2011*) resecretion, both depending on the endosomal sorting complex required for transport (ESCRT). Pools of Hh and ESCRT proteins are then secreted together into the extracellular space (*Gradilla et al., 2014*; *Matusek et al., 2014*; *Vyas et al., 2014*), Hh being transported on (*Bischoff et al., 2013*) or inside of (*Chen et al., 2017*) basolateral cytonemes, or apically released to promote Hh long-range activity (*Ayers et al., 2010*) (*Figure 12*). While Hh shedding may target several of these pools, timely and reliable paracrine Hh function through proteolytic release and extracellular apical diffusion alone (*Figure 12b''*) is difficult to envision for two reasons. First, patterning of folded epithelia, such as the *Drosophila* imaginal disc, poses a problem if spreading were to occur out of the plane of the epithelial cell layer through diffusion or flow, as this would result in morphogen loss into the peripodial space and loss of long-range Hh function. The second limitation is that it normally takes much time for diffusing molecules to travel long distances away from the source because the timescale of diffusion increases with the square of the distance (*Berg, 1993*; *Müller and Schier, 2011*). Cytoneme- or exosome-mediated basolateral transport, followed by proteolytic Hh relay over short distances at membrane contact sites (*González-Méndez et al., 2017*), would effectively solve both problems, as would the idea of heparan sulfate proteoglycan 'restricted' Hh transport at the apical cell surface (*Han et al., 2004*). Our future aim is to distinguish between these possibilities. We also aim to characterize the Hh release factor Shifted (*Glise et al., 2005*), a soluble protein with structural similarities to vertebrate Scube2 sheddase enhancers (*Jakobs et al., 2014*; *Jakobs et al., 2016*; *Jakobs et al., 2017*), to identify the elusive 'Hh sheddase'. Finally, we are currently investigating the important question of whether C-terminal Hh processing contributes to its in vivo biofunction in the wing disc and in other developing tissues requiring Hh signaling over shorter ranges, such as in the developing eye (*Ma et al., 1993*).

In conclusion, we propose that palmitate-controlled quantitative Hh shedding from the cell surface constitutes an essential step in Hh transmission and high-threshold tissue patterning in vivo. While we have used *Drosophila* wing development to elucidate this molecular process, we expect our results to also be relevant to other Hh-dependent developmental programs and to Hh ligand-dependent cancer induction and progression (*Amakye et al., 2013*).

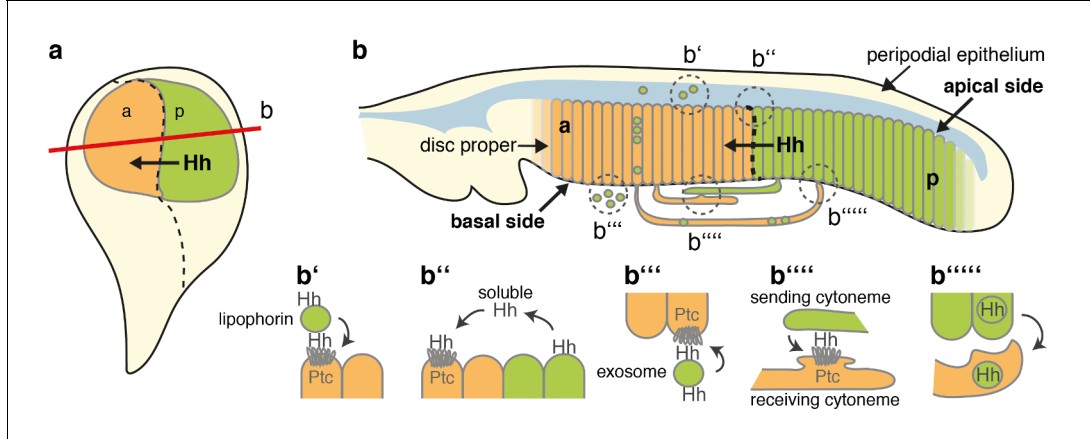

**Figure 12.** Model of membrane-dependent and membrane-independent Hh protein signaling from posterior producing cells to distant receiving cells and their potential congruency. (**a**) Schematic of a third instar wing disc with the Hh-producing posterior compartment labeled in green and the Ptc-receptor-producing anterior compartment labeled in orange. The a/p compartment border crossed by spreading Hh is shown as a dashed line. (**b**) Line drawing of a vertical section of the wing disc at a site marked with a red line in (**a**). Two pools of Hh are secreted from posterior columnar cells of the wing imaginal disc by different mechanisms (dashed circles). One pool is released from the apical side of the polar epithelium using lipophorins as hydrophilic carriers (**b'**) (*Panáková et al., 2005*) or via proteolytic processing of lipidated Hh membrane anchors, as suggested in this work (**b''**). Subsequently, unprocessed lipidated Hh or processed ectodomains diffuse through the fluid-filled peripodial space (labeled blue) to bind to Ptc receptors expressed in the anterior compartment. Another pool of apical cell-surface Hh is internalized and re-secreted apically (*D'Angelo et al., 2015*) or basolaterally (*Callejo et al., 2011*) using exosomes (**b'''**) (*Matusek et al., 2014*) or long cellular protrusions, known as cytonemes, as carriers. Cytonemes can extend from posterior Hh-producing cells to deliver Hh to cell surface receptors on receiving anterior cells in their close vicinity, or can meet 'receiving' cytonemes extending from more distant anterior cells at defined contact sites (**b''''**) (*González-Méndez et al., 2017*). Receiving anterior cytonemes that take up Hh from basal subcellular sites of expressing posterior cells for subsequent intracellular transport to the apical pole of anterior epithelial cells have also been described (**b'''''**) (*Chen et al., 2017*). However, an explanation is needed about how lipidated Hh can 'switch' between sending and receiving cytonemes, or relay from vesicular or cytoneme membranes to their receptors. This problem may be solved by proteolytic Hh processing, resulting in its relay between cytonemes or from producing cell membranes to Ptc receptors on receiving anterior cells. We note that the findings presented in this work do not support the alternative possibility, that is, that different transport modes work in parallel, because HA insertion into the N-terminal Hh processing site abolished (most) Hh-dependent patterning of the L3-L4 intervein region in a dominant-negative manner.

DOI: https://doi.org/10.7554/eLife.33033.022

# Materials and methods

### Key resources table

| Reagent type (species) or resource | Designation | Source or reference | Identifiers | Additional information |
|---|---|---|---|---|
| Gene (*Drosophila melanogaster*) | Hedgehog; Hh | PMID 8252628 | NCBI Reference sequence: NM_001038976.1 | |
| Cell line (*D. melanogaster*) | Schneider 2 | Invitrogen | RRID: CVCL_Z232 | |
| Cell line (*Homo sapiens*) | Bosc23 | PMID: 11395778 | RRID: CVCL_4401 | |
| Transfected construct (Hedgehog) | Hh | PMID 8252628 | NCBI Reference sequence: NM_001038976 | |
| Transfected construct (Sonic hedgehog) | Shh | PMID: 7916661 | NCBI Reference sequence: NM_009170 | |
| Antibody | anti-Hh | Santa Cruz | d300 catalog # sc-25759 | 2000-fold at 4°C over night |
| Antibody | anti-Shh | R and D Systems | Catalog # AF464 | 1000-fold at 4°C over night |
| Antibody | anti-en | DSHB | DSHB # 4D9 | 50-fold at 4°C over night |
| Antibody | anti-LacZ | Cappel, MP Biomedicals | Catalog # 08559761 | 50-fold at 4°C over night |
| Antibody | anti-HA | Sigma | catalog # H9658 | 5000-fold at 4°C over night |

*Continued on next page*

*Continued*

| Reagent type (species) or resource | Designation | Source or reference | Identifiers | Additional information |
|---|---|---|---|---|
| Strain, strain background (*D. melanogaster*) | Ptc > Gal4 | Bloomington # 2017 | | |
| Strain, strain background (*D. melanogaster*) | En > Gal4 | FlyBaseID FBrf0098595 | | |
| Strain, strain background (*D. melanogaster*) | Hh > Gal4 | Bloomington # 67046 | | |
| Strain, strain background (*D. melanogaster*) | 34B > Gal4 | Bloomington # 1967 | | |

## Fly lines

The following fly lines were used: *Ptc-Gal4* (*ptc>*): *w[\*]; P(w[+mW.hs]=GawB)ptc[559.1]*, Bloomington stock #2017; *En-Gal4e16E* (*En>*): *P(en2.4-GAL4)e16E*, FlyBaseID FBrf0098595; *Hh-Gal4* (*hh>*): *w[\*];; P(w[+mC]=Gal4)hh[Gal4]*, Bloomington stock #67046; *en(2)-Gal4* (*en(2)>*): *w[1118];; P(GMR94D09-Gal4)*, Bloomington stock #48011; *34B-Gal4* (*34B>*): *y[1]w[\*];; P(w[+mW.hs]=GawB)34B*, Bloomington stock #1967. These lines were crossed with flies homozygous for *UAS-hh* or variants thereof. All Hh cDNAs cloned into pUAST-attP were first expressed in *Drosophila* S2 cells to confirm correct protein processing and secretion. Transgenic flies were generated by using the landing site *51* C1 by Best-Gene or in-house by using strain *PhiC31(X); attPVK37; attP2* that possesses the landing sites *VK37* and *attP2*. Cassette exchange was mediated by germ-line-specific phiC31 integrase (*Bateman et al., 2006*). *Ptc-LacZ* reporter flies were kindly provided by Jianhang Jia, Markey Cancer Center, and Department of Molecular and Cellular Biochemistry, University of Kentucky College of Medicine, Lexington, USA.

## Confocal microscopy

Wing discs were fixed, permeabilized and stained with anti-β-galactosidase antibodies (Cappel, MP Biomedicals) and Cy3-conjugated goat-α-rabbit antibodies (Jackson Immuno Research). Posterior, Hh-producing cells were detected with monoclonal antibodies directed against engrailed (en 4D9, DSHB) and Alexa488-conjugated donkey-α-mouse antibodies (Thermo Fisher). Images were taken on a LSM 700 Zeiss confocal microscope using ZEN software. Maximum intensity projections are shown.

## Cloning and expression of recombinant proteins

Hh cDNA (nucleotides 1–1416, corresponding to amino acids 1–471 of *D. melanogaster* Hh) and HhN cDNA (nucleotides 1–771, corresponding to amino acids 1–257) were inserted into pENTR, sequenced, and cloned into pUAST for protein expression in S2 cells or the generation of transgenic flies. Mutations were introduced by QuickChange Lightning site-directed mutagenesis (Stratagene). Primer sequences and sequence information is shown in *Supplementary file 1*. S2 cells (RRID: CVCL_Z232) were cultured in Schneider's medium (Invitrogen) supplemented with 10% fetal calf serum (FCS) and 100 µg/ml penicillin/streptomycin. The cells were obtained from C. Klämbt, University of Münster, Germany, and tested negative for mycoplasma. S2 cells were transfected with constructs encoding Hh and HhN variants together with a vector encoding an actin-Gal4 driver by using Effectene (Qiagen) and cultured for 48 hr in Schneider's medium before protein was harvested from the supernatant. Shh constructs were generated from murine cDNA (NM_009170) by PCR (primers are listed in *Supplementary file 1*). Hh acyltransferase cDNA (NM_018194) was obtained from ImaGenes and cloned into pIRES (ClonTech) for bicistronic Shh/Hh acyltransferase coexpression in the same transfected cells. This resulted in N-palmitoylated, C-cholesterylated proteins. Bosc23 cells (RRID: CVCL_4401) were cultured in Dulbecco's modified Eagle's medium (Lonza) supplemented with 10% FCS and 100 µg/ml penicillin-streptomycin. The cells were obtained from D. Robbins, University of Miami, USA, authenticated via by PCR-single-locus-technology (Eurofins Forensics), and tested negative for mycoplasma. Bosc23 cells were transfected with PolyFect (Quiagen) and cultured for 48 hr, the medium was changed, and Shh was secreted into serum-free medium for the indicated times. All media were ultracentrifuged for 30 min at 125,000 g, and the proteins were TCA

precipitated and analyzed by 15% SDS-PAGE and western blotting with polyvinylidene difluoride membranes. Blotted proteins were detected by α-HA antibodies (mouse IgG; Sigma), α-Shh antibodies (goat IgG; R and D Systems), or α-Hh (rabbit IgG, Santa Cruz Biotechnology). Incubation with peroxidase-conjugated donkey-α-goat/rabbit/mouse IgG (Dianova) was followed by chemiluminescent detection (Pierce). Photoshop was used to convert grayscale blots into merged RGB pictures for improved visualization of terminal peptide processing.

## Preparation of Drosophila larval lysates

*Drosophila* third-instar larvae were collected and transferred into a microcentrifuge to which 1 ml lysis buffer was added (PBS containing 1% (v/v) Triton X-100). Larvae were homogenized with a micropestle and the solution was cleared at 15,000 rpm for 15 min at 4℃. The supernatant was sterile-filtered (45 μm) and transferred into a fresh microcentrifuge tube for gel filtration analysis. All processings were conducted at 4℃.

## Chromatography

Gel filtration analysis was performed on an Äkta protein purifier (GE Healthcare) on a Superdex200 10/300 GL column (Pharmacia) equilibrated with PBS at 4℃. Eluted fractions were TCA precipitated and analyzed by SDS-PAGE as described earlier. Signals were quantified by using ImageJ.

## Bioanalytical and statistical analysis

Sequence analysis was conducted on the CFSSP secondary structure prediction server (http://www.biogem.org/tool/chou-fasman/). All statistical analyses were performed in GraphPad Prism by using the Student's *t* test (two-tailed, unpaired, confidence interval 95%). For wing quantifications, 10 male and 10 female wings were analyzed for each data set and ratios between L3-L4 intervein areas and L2-L3 intervein areas determined. All error estimates are standard errors of the mean (SEM).

## Immunoelectron microscopy

Shh-expressing Bosc23 cells were fixed overnight at 4℃ in 4% paraformaldehyde/glutaraldehyde, washed in PIPES, and dehydrated in a graded ethanol series (30% EtOH, 4℃, 45 min; 50% EtOH, −20℃, 1 hr; 70% EtOH, −20℃, 1 hr; 90% EtOH, −20℃, 1.5 hr; 100% EtOH, −20℃, 1.5 hr; 100% EtOH, −20℃, 1.5 hr). Dehydrated cells were embedded in Lowicryl K4M embedding medium by using the Lowicryl K4M Polar Kit (Polysciences). Cells were then embedded in gelatin capsules, centrifuged twice for 15 min at 1500 rpm, and incubated overnight at −35℃. For polymerization, the resin was UV irradiated for 2 days at −35℃. The embedded samples were cut into 60 nm sections, washed in PBS containing 5% BSA (pH 7.4), and incubated for 2 hr in PBS-BSA containing primary antibodies (α-Shh antibodies from R and D, GeneTex, and Cell Signaling at 1:20 dilution). Samples were washed five times in PBS-BSA and once in Tris-BSA. Secondary antibodies conjugated to 5 nm and 10 nm gold nanoparticles were diluted in Tris-BSA buffer and incubated with the cell sections for 1 hr. Afterwards, samples were washed five times in Tris-BSA and once in dH$_2$O. Contrasting was done with 2% uranyl acetate (15 min) and Reynold's lead citrate (3 min). Finally, immunogold-labeled cell sections were analyzed by using a transmission electron microscope (CM10, Philips Electron Optics).

## Data availability

The transgenic fly lines Hh-CW (lacking the putative N-terminal Hh processing site), Hh-CW/HA (a variant having this site replaced with a hemagglutinin tag) and HhHS (carrying a C-terminally inserted HA-tag) generated in the course of this study that support the phenotypes described in the manuscript are available upon request from the corresponding author (KG). We plan to publish these new lines separately in the future.

## Acknowledgements

The excellent technical and organizational assistance of S Kupich and R Schulz is gratefully acknowledged. The authors thank Marius Mählen for contributing to this work. This work was financed by

DFG (German Research Council) GRK1549/1, GR1748/4-1, GR1748/5-1, and CiM FF-2015–02 support.

## Additional information

### Funding

| Funder | Grant reference number | Author |
| --- | --- | --- |
| Deutsche Forschungsgemeinschaft | GRK1549/1 | Kay Grobe |
| Deutsche Forschungsgemeinschaft | GR1748/4-1 | Kay Grobe |
| Deutsche Forschungsgemeinschaft | GR1748/5-1 | Kay Grobe |
| Cells-in-Motion Cluster of Excellence | FF-2015-02 | Milos Galic Kay Grobe |

The funders had no role in study design, data collection and interpretation, or the decision to submit the work for publication.

### Author contributions

Sabine Schürmann, Conceptualization, Investigation, Writing—original draft; Georg Steffes, Conceptualization, Supervision, Investigation; Dominique Manikowski, Philipp Kastl, Shyam Bandari, Stefanie Ohlig, Corinna Ortmann, Investigation; Ursula Malkus, Conceptualization, Supervision, Validation, Investigation; Rocio Rebollido-Rios, Mandy Otto, Investigation, Visualization; Harald Nüsse, Supervision, Investigation, Methodology; Daniel Hoffmann, Christian Klämbt, Conceptualization, Supervision, Methodology, Writing—review and editing; Milos Galic, Conceptualization, Resources, Supervision, Methodology, Writing—review and editing; Jürgen Klingauf, Conceptualization, Resources; Kay Grobe, Conceptualization, Resources, Supervision, Funding acquisition, Validation, Writing—original draft, Project administration, Writing—review and editing

### Author ORCIDs

Daniel Hoffmann ⓘD https://orcid.org/0000-0003-2973-7869
Kay Grobe ⓘD https://orcid.org/0000-0002-8385-5877

### Decision letter and Author response

Decision letter https://doi.org/10.7554/eLife.33033.029
Author response https://doi.org/10.7554/eLife.33033.030

## Additional files

### Supplementary files

• Supplementary file 1. Mutagenesis primers and sequence confirmation for all Hh variants used in this study.
DOI: https://doi.org/10.7554/eLife.33033.023

### Major datasets

The following dataset was generated:

| Author(s) | Year | Dataset title | Dataset URL | Database, license, and accessibility information |
|---|---|---|---|---|
| Schuermann S, Steffes G, Manikowski D, Kastl P, Malkus U, Bandari S, Ohlig S, Ortmann C, Rebollido-Rios R, Otto M, Nuesse H, Hoffmann D, Klaembt C, Galic M, Klingauf J, Grobe K | 2018 | Data from: Proteolytic processing of palmitoylated Hedgehog peptides specifies the 3-4 intervein region of the Drosophila wing | http://dx.doi.org/10.5061/dryad.6b058n7 | Available at Dryad Digital Repository under a CC0 Public Domain Dedication |

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
