## [Decision Letter]

[Editors’ note: a previous version of this study was rejected after peer review, but the authors submitted for reconsideration. The first decision letter after peer review is shown below.]

Thank you for submitting your work entitled "Proteolytic processing of palmitoylated Hedgehog peptides specifies the 3-4 intervein region of the *Drosophila* wing" for consideration by *eLife*. Your article has been reviewed by a Senior Editor and three reviewers, one of whom is a member of our Board of Reviewing Editors.

Our decision has been reached after consultation between the reviewers. Based on these discussions and the individual reviews below, we regret to inform you that your work will not be considered further for publication in *eLife*.

All the reviewers found the paper to be potentially interesting: it is well written and deals with an important topic. But after discussion the consensus view was that the conclusions are not yet well enough supported by the experimental data to justify publication in *eLife*. In addition to the need for the experiments described to be more rigorously performed, the proposed model makes several predictions that could be tested to provide further evidence.

The reviewers were keen to point out that although *eLife* has a policy of rejecting papers where it is judged that more than 2 months will be needed for revisions, were this work to develop into a more mature story, there is no barrier to resubmitting it in the future.

*Reviewer #1:*

This is an interesting paper that uses a combination of microscopy, cell biology and *Drosophila* genetics to carefully dissect the mechanisms that underlie Hh release and intercellular trafficking. The aim is to resolve the mystery of how a dual lipidated extracellular protein, which oligomerizes and tightly associates with the cell surface of the producing cell, is nevertheless transported across an epithelial field of cells to act on cells at a distance. Several models have been proposed, including trafficking by exosomes or cytonemes, but the conclusion of this paper is that proteolytic shedding of the palmitoylated N-terminal of Hh is the key step.

This is a thought-provoking and carefully designed set of experiments that lead to some plausible and generally well supported conclusions. It's also an important topic, being a central issue about the regulation of a primary signaling pathway of relevance to development and disease. In general, I think it is appropriate for a journal like *eLife*.

It is, however, a complicated story, and one which still leaves quite a few questions: it will not be the last word on this topic.

Although the experiments are well planned and quite elegant, I found some are hard to follow. I'd suggest more use of diagrams to outline the logic of individual experiments. An example would be the use of Ptc>Hh^C85S^ to discriminate between autonomous and non-autonomous effects. People not used to fly developmental biology will find this difficult. Another case would be the Bosc23 cells expressing the differently tagged versions.

The tone of many of the results is quite definitive and I am not sure this is wholly justified. The results are clear and the interpretations plausible, but I am not convinced that in each case their interpretation is the only way of interpreting the data. The paper would be stronger if the current results were more clearly set in the context of other models, with more discussion about how these results challenge or support different views. Since there are already quite well discussed models of Hh trafficking, the current authors need to leave the reader with as clear as possible a view of what this new paper changes and what are the still outstanding questions.

One gap I detected was any discussion about the C-terminal cleavage. If I understand their model correctly, this is also essential for release of active Hh. Is the view that the clustering is also necessary for that? And do they envisage that the N- and C-terminal cleavages are done by the same or different proteases? Moreover, what is known about the identity of the proteases should also be discussed.

*Reviewer #2:*

The maturation of Sonic Hedgehog (Shh) is very atypical as the final mature protein is dually lipid modified, by an N-term palmitic acid and a covalent link with a molecule of cholesterol on the C-terminus. How it is extracted from the plasma membrane bilayer and is transported to long-range targets is still enigmatic.

The laboratory of Kay Grobe has previously published that SHH is secreted as a multimer in which the lipidated N-term peptide inhibits each adjacent SHH molecule by masking its receptor binding site. The lab has proposed that SHH dual lipids are shed at the surface of producing cells, allowing SHH propagation and activation. The cleavage of these inhibitory peptides therefore unmasks receptor-binding sites of all SHH molecules present in the multimer and permits its propagation and activation of signaling (Ohlig et al., 2011). This last study was performed using in most cases the human kidney cell line HEK293 and its derivative the Bosc23 cell line, which is a model for cancer research. In this new manuscript, the authors use *Drosophila* transgenics to confirm their model. As appropriate controls are missing, I am overall not convinced about the accuracy of the conclusions proposed here. I highlight below my main critics:

Overexpression of the non-palmitoyled HH variant in wild type HH producing cells of the *Drosophila* wing discs reveals a dominant negative activity on the final adult wing pattern. The overexpression of this variant in HH receiving cells had only a weak effect. From this difference the authors came to the conclusion that the non-palmitoyled HH variant display a dominant negative effect only when expressed in the same cells with the wild type protein and forming a mixed cluster. This is the basis of the model but I believe several other hypothesis can be proposed (in addition, the fact that not HH target genes are presented in this study weaken the entire picture). For example, the authors do not show that the two drivers express the HH variant at the same level which could explain a differential dominant negative effect. It is also possible that upon forced expression of HH in anterior cells, the protein is not secreted at the correct pole (apical/basal) of the cell and thus cannot bind properly to its receptor. No staining of extracellular HH is provided here. Conventional and extracellular staining of the different HH variants is imperative to get a clearer picture of the behavior of these variants in vivo. One should note that overexpression of this variant has been published previously by several other groups in 2006 and showed an opposite result, with expansion of target genes in both embryos and wing imaginal discs. This is not commented by the authors.

The authors showed that N-terminal truncation of HH reversed the dominant negative activity of the non-palmitoyled HH variant when expressed in flies. These new variants are not forming multimere but monomere. From this, the authors proposed that the inhibitory effect of unprocessed N-terminal peptides is linked to its presence in the multimere. But it is also possible that these new variants cannot bind properly Hh receptor and thus are just "neutral" with no activity whereas the non-palmitoyled variant binds the receptor but do not activate signaling properly. There are numerous in vitro assays that could be performed to test the author's hypothesis, for example, why not analyze binding affinity of HH variants to Ptc with synthesized peptides?

Altogether, the authors propose a model in which Hh is secreted to the plasma membrane as dually lipidated molecule, assembled as multimer at the cell surface, and then cleaved for propagation. What is surprising with the author's model is that both lipids need to be cleaved to release Hh functional cluster from producing cells. This is intriguing as numerous labs have shown that expression of a HH variant without lipid is secreted as monomere with no activity. So, the authors would need to resolve two important questions to confirm their model. First, how the cleaved HH molecule is kept assembled as multimer, and second, how the putative multimer of non-lipidated HH proposed here is able to provide signaling activation when a monomeric non-lipidated HH cannot.

In conclusion, I did not find that the new data provide convincing data and progress for a firm evidence of the model proposed here. I also advise this team to take into more consideration many of the in vivo data already published by other labs, which in several cases appears to be contradictory to the author's own observation.

*Reviewer #3:*

The manuscript by Grobe and colleagues use the *Drosophila* wing as the principal model system to analyse the functional role of protein palmitoylation in the N-terminal cysteine of Hedgehog (Hh) signalling molecule. Authors present evidence that mutant forms of *Drosophila* Hh unable to be palmitoylated (Hh-C85S) or proteolytically processed (HA-Hh) gives rise to a Hh-loss of function adult wing phenotype when overexpressed in Hh-expressing but not responding cells, and that truncation of the N-terminal region of Hh-C85S or a mutation of the N-terminal cysteine in HA-Hh restore these phenotypes. Authors propose that palmitoylation of Hh contributes to the clustering of Hh molecules at the cell membrane of Hh-expressing cells and to the tightly controlled-proteolytic processing and release of an active form of Hh able to bind to Ptc in nearby cells.

The paper is excellently written and the proposed working model is consistent with the results. Authors have done an excellent and thorough exercise by using the *Drosophila* wing together with EM data and cell culture experiments and propose a mechanistic model of the biological function of Hh-palmitoylation.

Major issue:

Authors conclude that the adult wing phenotype caused by expression of Hh-C85S or HA-Hh in the P compartment is caused by a dominant negative effect on the endogenous Hh molecule (as nicely depicted in their cartoons). Authors should present evidence (1) that the wing phenotype can be rescued by co-expression of a wild type form of Hh, (2) that release of a wild type form of Hh into the adjacent compartment is compromised by Hh-C85S or HA-Hh co-expression (by using available UAS-Hh-GFP transgenic flies), and (3) that anterior cells abutting the P compartment are still sensing Hh signalling (similarly to a tethered form of Hh). Thus, expression of a wild type form of Hh and Hh target genes should be analysed in wing primordia of all genotypes (en-gal4; UAS-Hh-C85S +/- UAS-Hh-GFP and en-gal4; UAS-Hh-A +/- UAS-Hh-GFP) to confirm their proposal. These experiments are feasible.

Other issues:

1) Were male of female wings used in their L3-L4/L2-L3 region measurements?

2) Authors should notice that en-gal4 is also expressed in the A compartment, although at lower levels and later in development. The ability of Hh-C85S and HA-Hh to cause a wing phenotype should be analysed also with the hh-gal4 driver (again available in the fly community).

3) Expression of Hh-C85S in the ptc domain has not phenotype (Figure 3H) while Hh-HA does Figure 6H, I). Why?

4) How certain are authors that veins in Figure 6B are L4 and L5 and not L2 and L5. The latter would be expected as these are the ones that depend on Dpp signalling.

[Editors’ note: what now follows is the decision letter after the authors submitted for further consideration.]

Thank you for resubmitting your work entitled "Proteolytic processing of palmitoylated Hedgehog peptides specifies the 3-4 intervein region of the *Drosophila* wing" for further consideration at *eLife*. Your revised article has been favorably evaluated by Naama Barkai (Senior Editor), a Reviewing Editor, and three reviewers.

The manuscript has been improved but there are some remaining issues that need to be addressed before acceptance, as outlined below.

The editors and reviewers have discussed your paper extensively. Our position is that your work is interesting, provocative and of potential broad significance. On the other hand, there is a clear consensus that you do not currently provide a clear enough description of the controversy and uncertainty that surrounds your interpretation. Your paper does not yet allow a definitive conclusion about the model.

*eLife* is an appropriate journal for this kind of work, but only if the authors are able clearly and explicitly to describe their work in the wider context, allowing a reader to form a balanced picture of where the field stands. We would therefore like to give you one more opportunity to respond to remaining concerns but also – and even more importantly – to rewrite the Introduction and Discussion in a way that gives that broader context and describes to a non-expert reader the uncertainties that remain about your model. What would be a definitive experiment that might prove or refute it? Which areas are you least confident about? Do you have a plan about how to address them? How do you reconcile your ideas with previous apparently contradictory data?

Science is full of uncertainty and journals like *eLife* should not encourage this to be hidden behind unrealistic confidence or advocacy. If you can amend your paper to provide a fair and balanced description of how the field stands, and how your new data move it forward, but also where the uncertainties still lie, we would be happy to publish it. If you feel that this is not in your best interest and you prefer to argue for your model more directly, our decision would be not to accept it.

In addition to these general points there are still some remaining specific issues in the reviews below. We would encourage you to address these as fully as possible. Where necessary and appropriate, please acknowledge if they provide significant challenges to your model.

*Reviewer #1:*

As it is the second time I review this work, I will directly follow the main comments I have put together for the first manuscript.

My previous comments: "Overexpression of the non-palmitoyled HH variant in wild type HH producing cells of the *Drosophila* wing discs reveals a dominant negative activity on the final adult wing pattern. The overexpression of this variant in HH receiving cells had only a weak effect. From this difference the authors came to the conclusion that the non-palmitoyled HH variant display a dominant negative effect only when expressed in the same cells with the wild type protein and forming a mixed cluster. This is the basis of the model but I believe several other hypothesis can be proposed (in addition, the fact that not HH target genes are presented in this study weaken the entire picture).”

The authors made some effort (by providing target gene expression) to answer this first comment. Nevertheless, there are numerous examples in the literature showing that expression of the non-palmitoyled HH variant in the anterior cells of the wing disc leads to a strong dominant negative effect (see for example, Figure D in Lee et al., 2001, Treissman lab). In the same paper, the authors mentioned that overexpression of the non-palmitoyled SHH variant in the posterior region of the mouse limb have no inhibitory effect on endogenous Shh activity, which contradict the model proposed here.

So, the first proposal presented here, that, in producing cells, non-palmitoyled HH proteins associate in cluster with other HH protein in order to act as a dominant negative variant does not fit with published data. In addition, if this model was correct, one would expect a stoichiometric dominant negative effect of the non-palmitoyled HH variant, which again is not supported by the study of Williams et al., 1999.

Also, I still do not understand the model based on a "model linear *Drosophila* Hh cluster", which I assume is based on the published crystallographic structure of vertebrate Shh. Surprisingly, a non-palmitoyled SHH does not form multimers, whereas a non-palmitoyled *Drosophila* HH does so (Figure 7 in Chen et al., 2004, Pao-Tien Chuang lab). So it is not clear to me how one can use information from Shh and extrapolate to Hh behavior, knowing that these two proteins behave differently.

My previous comments: "No staining of extracellular HH is provided here. Conventional and extracellular staining of the different HH variants is imperative to get a clearer picture of the behavior of these variants in vivo."

Authors: “As has been pointed out by Art Landers group in Irvine, reliable fluorescence microscopic visualization of free extracellular protein transport is close to impossible:

Using Dpp as an example, it was estimated that free extracellular protein is unlikely to account for more than 3% of total morphogen, and less than 1% of what is normally visualized by fluorescence imaging (Zhou et al., 2012). […] For all of these reasons, we refrained from trying to visualize extracellular Hh in vivo by fluorescence microscopy, and decided to use clear unambiguous genetic assays instead.”

I still believe that, a non-detergent staining at 4 °C with Hh and/or HA antibodies will provide information such as: are these Hh variants accumulating at the cell surface of producing cells, how do they distribute in anterior cells compared to wt Hh etc.? There are also tricks (with for example the expression of a Ptc variant which is not internalized) to increase and visualize accumulation of Hh at the cell surface of receiving cells. As previous studies demonstrated that Ptc-binding restricts Hh spreading, one would expect that HhC25S-containing Hh multimers would spread more anteriorly because they bind Ptc less well compared with wild type Hh as suggested by the present model. Also, there are Hh variants presented in this study which are likely not binding to Ptc, such as the HHC85S;delta86-100. Indeed, a deletion of the first 20 NH2 terminal residues of SHH leads to a protein that do not bind to Ptc anymore (Williams et al., 1999).

My previous comment: “One should note that overexpression of this variant has been published previously by several other groups in 2006 and showed an opposite result, with expansion of target genes in both embryos and wing imaginal discs. This is not commented by the authors.”

Authors: “We wonder which papers and which variant the reviewer does refer to. If it is HhC85S, we are not aware of any publication claiming that this form has any bioactivity; the general consensus is that non-palmitoylated invertebrate Hh is always completely inactive.”

Well, both studies described in Gallet et al., 2003 (Figure 2) and Callejo et al., 2006 (Figure 6) showed that embryonic wg, and larval dpp and iro expressions are expanded under overexpression of HhC85S.

Another issue which needs clarification regards the putative multimer of non-lipidated HH which is proposed in the model here (Figure 5A). How is this multimer able to provide signaling activation whereas a monomeric non-lipidated HH cannot? And how the Hh proteins is kept assembled in the multimer without lipids? This is difficult to understand for me. The authors might be right but their model of cell-surface shedding relies mainly on in vitro experiment in which Shh is expressed at a non-physiological level in cultured cells. So far, if I am correct, in all their "Hh shedding" studies, the authors used the human kidney cell line HEK293 and its derivative the Bosc23 cell line, which is a model for cancer research and which might not mimic physiological conditions.

Interestingly, the acyl transferase that catalyzes the transfer of a palmitoyl moiety to Hh has been identified in flies. When its homolog is mutated in mice (Chen et al., 2004), digit 3 to 5 (which depends on long range activity of Shh) still form, suggesting that, if Shh assembles in multimere in this mutant, the non-palmitoyled SHH is still able to bind Ptc and activate the pathway. How do the authors reconcile their data with this observation?

Furthermore, a wide variety of hydrophobic modifications have been shown to increase the potency of Shh when added at the N-terminus of the protein, ranging from long-chain fatty acids to hydrophobic amino acids (Taylor et al., 2001). I do not see how this fits with the authors model.

Overall, this is a story that is ultimately still not convincing for this reviewer. The model proposed here cannot explain many of the in vivo observations obtained by other labs, some of which are listed above.

*Reviewer #2:*

Schuermann et al. provide evidence that Hh activity and spread depend on the proteolytic removal of the palmitoylated N-terminal peptide. The paper contains a huge amount of data but there is no "killer experiment" that definitively proves the proposed model. For example, most of the conclusions rest on the use of ectopic expression of mutant versions in a wild-type background instead of replacement of the endogenous hh gene. Moreover, no in vivo localization data is provided. I therefore encourage the authors to raise the possibility that there might be other interpretations of the data and discuss the need for follow up studies of Hh versions expressed from the endogenous locus and visualized in vivo. Despite these concerns, the findings are interesting, provocative and of broad interest.

Other concerns:

1) Is the packing seen in Figure 1 really due to Hh or could it be caused by the confirmation and size of the antibodies?

2) The evidence that the mutant versions are expressed at similar levels as wild-type Hh is sparse.

3) The authors try to build models on complex regulatory loops (e.g. ptc feedback in Figure 2) but the outcomes of such loops are very difficult to predict.

4) Subsection “N-palmitate controls Hh release from the cell surface in vitro”, last paragraph: "compelled"?

*Reviewer #3:*

Authors have successfully addressed all my concerns, so I believe the manuscript is ready for *eLife*.

---

## [Author Response]

[Editors’ note: the author responses to the first round of peer review follow.]

Reviewer #1:This is an interesting paper that uses a combination of microscopy, cell biology and Drosophila genetics to carefully dissect the mechanisms that underlie Hh release and intercellular trafficking. The aim is to resolve the mystery of how a dual lipidated extracellular protein, which oligomerizes and tightly associates with the cell surface of the producing cell, is nevertheless transported across an epithelial field of cells to act on cells at a distance. Several models have been proposed, including trafficking by exosomes or cytonemes, but the conclusion of this paper is that proteolytic shedding of the palmitoylated N-terminal of Hh is the key step.This is a thought-provoking and carefully designed set of experiments that lead to some plausible and generally well supported conclusions. It's also an important topic, being a central issue about the regulation of a primary signaling pathway of relevance to development and disease. In general, I think it is appropriate for a journal like eLife.It is, however, a complicated story, and one which still leaves quite a few questions: it will not be the last word on this topic.Although the experiments are well planned and quite elegant, I found some are hard to follow. I'd suggest more use of diagrams to outline the logic of individual experiments. An example would be the use of Ptc>Hh^C85S^ to discriminate between autonomous and non-autonomous effects. People not used to fly developmental biology will find this difficult. Another case would be the Bosc23 cells expressing the differently tagged versions.

To improve the manuscript according to this suggestion, illustrations of results or schematics of Hh variants have been added: Figure 2 now shows wing disc stainings for engrailed (expressed in the Hh-producing posterior compartment) and the non-overlapping (anterior) ptc-expression domain to illustrate spatially connected / disconnected transgene expression. Figure 3 is added to the manuscript to confirm that Hh co-clustering constitutes the basis for dominant-negative activity of non-palmitoylated Hh (this is also illustrated). Finally, Figure 6 now contains schematics of the various Hh variants used including their detected cleavage sites.

The tone of many of the results is quite definitive and I am not sure this is wholly justified. The results are clear and the interpretations plausible, but I am not convinced that in each case their interpretation is the only way of interpreting the data. The paper would be stronger if the current results were more clearly set in the context of other models, with more discussion about how these results challenge or support different views. Since there are already quite well discussed models of Hh trafficking, the current authors need to leave the reader with as clear as possible a view of what this new paper changes and what are the still outstanding questions.

We agree with this reviewer that our conclusions sound strong, and we now toned them down and provided additional background, as requested. However, we suggest some words of caution should remain stated in the manuscript, especially that the cloning protocol for generating tagged, membrane tethered Hh-GFP from endosomes/exosomes/cytonemes (1-6) exactly matches a protocol to generate permanent GFP membrane markers (no Hh involved!) for the fluorescence microscopic visualization of cell membranes (7). This fact is important for the correct interpretation of published models that mostly derive from the microscopic analysis of Hh-GFP subcellular localization.

In addition to that, data from our lab indicates (please refer to our responses to concerns raised by reviewer 2,) that Hh-GFP used in the above publications undergo almost quantitative proteolytic processing in L3 discs in vivo. This proteolytic processing results in a major fraction of cholesteroylated GFP and free (bioactive) Hh. This adds another critical problem to the above models: while the subcellular route of (irreversibly membrane tethered and non-physiological) GFP was followed by fluorescence microscopy, released Hh may have taken an entirely different route that escaped visualization.

Moreover, we observed that Hh^C85S^-GFP does not behave like untagged Hh^C85S^ in vivo, indicating additional problems associated with the tag. Despite these observations, which make us now even more sceptical of models derived from the study of Hh-GFP, and which make it hard for us to draw a useful and honest connection between these models and our data, we added a paragraph on the relevance of our results for these models (Discussion section).

Finally, we would like to state that additional ongoing studies of flies carrying various deletions, mutations and insertions into the N-terminal Hh target site all recapitulate the data shown in this manuscript, and that electron microscopic analysis detected Hh clusters not associated with the cell membrane in fly wing discs in vivo, indicating their solubilisation. These data can be provided upon request.

One gap I detected was any discussion about the C-terminal cleavage. If I understand their model correctly, this is also essential for release of active Hh. Is the view that the clustering is also necessary for that? And do they envisage that the N- and C-terminal cleavages are done by the same or different proteases? Moreover, what is known about the identity of the proteases should also be discussed.

We indeed also study flies carrying deletions, insertions and mutations in the C- terminal Hh stem region (Four independent constructs, plus Hh-GFP used and published by others as mentioned above). In most cases, these mutations abolish all or most Hh bioactivity in vivo, as expected. This is consistent with published data demonstrating that non-steroylated Hh is present and highly active in vitro and in vivo (8). We also have indications that N- and C-terminal cleavages depend on different proteases, and (based on inhibitor studies) that Furin or related proteases target the N-terminal polybasic Hh Cardin-Weintraub-motif. These studies are all in progress, and we therefore ask that none of the preliminary data are to be included into the current manuscript.

Reviewer #2:[…] This is the basis of the model but I believe several other hypothesis can be proposed (in addition, the fact that not HH target genes are presented in this study weaken the entire picture).

In the revised manuscript, we provide en, ptc and dpp target gene expression for palmitoylated and non-palmitoylated Hh variants that is consistent with observed wing phenotypes (Figures 8 and 10) and the mechanism that we propose. Dominant-negative HAHh eliminates (almost) all anterior Hh-regulated en- and ptc-expression, as suggested by wing phenotypes lacking all L3/L4 intervein tissue, while ^HA^Hh^C85S^ represses target gene expression much less, if compared to HAHh and wild-type discs. This is consistent with the model that HAHh prevents the release of associated Hh from the producing cell surface. Moreover, we now also provide rescue data to confirm our model (please see comments to reviewer 3).

From this difference the authors came to the conclusion that the non-palmitoyled HH variant display a dominant negative effect only when expressed in the same cells with the wild type protein and forming a mixed cluster […] I believe several other hypothesis can be proposed […] For example, the authors do not show that the two drivers express the HH variant at the same level which could explain a differential dominant negative effect.

The reviewer is correct in stating that most different driver lines will express transgenes to different levels. Indeed, this is to be expected. In reply, we would like to point out the following:

First, in our first set of experiments, only one (the same) driver line is used for different Hh variants (cell-tethered *versus* soluble Hh (Figure 3), non-truncated *versus* N- truncated Hh (Figure 4). The reviewer’s criticism thus does not apply to these experiments.

In our second set of experiments, we indeed changed the expression domain for a given Hh variant (from posterior (en>) to a stripe in the anterior region (Ptc>)).

Therefore, to address this reviewers concern, we co-expressed both, active Hh and dominant negative ^HA^Hh and ^HA^Hh^C85S^, in the same compartment (using only one driver for the simultaneous expression of both constructs). As also described in the reply to reviewer 3, these experiments confirmed our model: Dominant negative ^HA^Hh^C85S^ activities under en-control were rescued by Hh co-expression (because more wt protein was now present in mixed clusters, increasing their activity), yet ^HA^Hh dominant negative activities could not be rescued (more Hh was expressed, and relative numbers of membrane tethers of any given cluster reduced, but remaining tethers still prevented release). Along the same line, using the 34B-Gal4 driver for targeted Hh expression in anterior cells, Hh gain-of-function (as indicated by a *Moonrat*-like phenotype (9)) was fully reversed by ^HA^Hh co-expression (by preventing Hh release).

Both experiments (Figure 9 in the revised manuscript) make clear that different driver activities are not the cause for the observed wing phenotypes.

It is also possible that upon forced expression of HH in anterior cells, the protein is not secreted at the correct pole (apical/basal) of the cell and thus cannot bind properly to its receptor.

The hh*^Moonrat^* allele causes ectopic expression of the *hh* gene in the anterior wing compartment, where normally *hh* is not expressed (9). The resulting phenotype varies between clear overgrowth of the (usually distal) part of the wing to slight disorganization of the wing margin and the addition of extra vein material. The Hh*^Moonrat^* phenotype demonstrates that bioactive Hh is produced by anterior cells, and 34B-Gal4 driven expression of UAS-Hh in the anterior field (as described above and now shown in the revised manuscript, Figure 9, leading to the same phenotype) confirms this. We can thus not confirm any unspecific effect of anterior Hh expression.

No staining of extracellular HH is provided here. Conventional and extracellular staining of the different HH variants is imperative to get a clearer picture of the behavior of these variants in vivo.

As has been pointed out by Art Landers group in Irvine, reliable fluorescence microscopic visualization of free extracellular protein transport is close to impossible: Using Dpp as an example, it was estimated that free extracellular protein is unlikely to account for more than 3% of total morphogen, and less than 1% of what is normally visualized by fluorescence imaging (10). Most of the material that could be visualized is either receptor/surface-associated or internalized. This fact rules out the use of fluorescence microscopy to visualize released Hh. We also decided against visualization of GFP-tagged forms of Hh, because 1) the freely diffusible fraction cannot be visualized, as outlined above, but only the immobile fraction. 2) In our hands, GFP-tagged Hh and Hh^C85S^ show extensive proteolytic processing and Hh^C85S^-GFP acts unlike untagged Hh^C85S^ in vivo. For all of these reasons, we refrained from trying to visualize extracellular Hh in vivo by fluorescence microscopy, and decided to use clear unambiguous genetic assays instead.

However, in another project, conventional immuno-electron microscopy of *Drosophila* wing discs was performed that allowed for the specific discrimination between membrane-associated Hh (expressed under en-Gal4 control, large clustered groups, red/white arrowheads) and extracellular yet matrix-associated Hh. The in vivo detection of matrix-associated extracellular protein distant from membranes strongly supports Hh release via shedding, and also support data of others (8). The figure is planned to be part of another manuscript; it is therefore not included in this submission.

One should note that overexpression of this variant has been published previously by several other groups in 2006 and showed an opposite result, with expansion of target genes in both embryos and wing imaginal discs. This is not commented by the authors.

We wonder which papers and which variant the reviewer does refer to. If it is Hh^C85S^, we are not aware of any publication claiming that this form has any bioactivity; the general consensus is that non-palmitoylated invertebrate Hh is always completely inactive. If the reviewer refers to non-palmitoylated yet (variably) bioactive Sonic Hh in vertebrates, we would like to refer to our recently published work on this topic (11).

Here, bioactivation of unpalmitoylated Shh is caused by Scube2-assisted N-terminal shedding, which for the first time explains the important yet variable role of N-palmitate for (vertebrate) Hh activity.

The authors showed that N-terminal truncation of HH reversed the dominant negative activity of the non-palmitoyled HH variant when expressed in flies. These new variants are not forming multimere but monomere.

This must be a misunderstanding. As shown in Figure 4—figure supplement 2 and Figure 6—figure supplement 1, all cholesteroylated Hh variants used in this study multimerize.

From this, the authors proposed that the inhibitory effect of unprocessed N-terminal peptides is linked to its presence in the multimere. But it is also possible that these new variants cannot bind properly Hh receptor and thus are just "neutral" with no activity whereas the non-palmitoyled variant binds the receptor but do not activate signaling properly. There are numerous in vitro assays that could be performed to test the author's hypothesis, for example, why not analyze binding affinity of HH variants to Ptc with synthesized peptides?

If we understand the comment correctly, the reviewer refers to the possibility of differential receptor binding/activation N-palmitoylated and non-palmitoylated Hh. However, we would like to point out that the design of all experiments shown in our submitted study is to assay the biofunction of the endogenous protein that is only indirectly affected by our transgenic forms without intrinsic activity – receptor binding of endogenous Hh, and not of our mutants, is therefore our read-out. The possibility of differential receptor binding/activation thus does not apply here. In our re-submitted manuscript, we extend these experiments to using bioactive transgenic Hh together with various palmitoylated and non-palmitoylated recombinant proteins expressed from posterior and ectopic anterior cells – again, this does not affect receiving cells, as it is always the endogenous / recombinant bioactive Hh that is assayed.

The possibility of non-productive “blockade” of receptors is also not the case, as shown by ptc>Hh^C85S^ expression shown in our work: Here, wing patterning is still unaffected. This is described in the original and resubmitted manuscript (Figures 2 and 3, and Author response image 2).

In contrast, impaired Hh binding to its receptor Ptc by unprocessed N-terminal peptides has extensively been demonstrated in the past (12-16), and has recently been confirmed by showing that proteolytic processing effectively truncates and activates non- palmitoylated inactive Sonic Hh (Author response image 1) (11). Palmitate or the N-peptide, therefore, are not directly required for signalling – palmitate merely controls that Sonic Hh clusters are completely N-processed during their release.

**Author response image 1. respfig1:** N-terminal Shh processing activates Shh. Center: Intermolecular interactions observed in the human Shh crystal structure (pdb:3M1N) (17). Analysis of crystal symmetry mates revealed that N-terminal peptides (amino-terminal amino acids 25-45, blue) wrap around the symmetry-related molecule (green) and interact with its zinc-coordination site, which is the Ptc receptor binding site. Zinc is shown as a black sphere, the yellow surface marks the Ptc-receptor binding site, and the putative sheddase target site (called CW motif) is marked in red. Proteolytic processing during release truncates the protein (bottom) and increases its electrophoretic mobility (left). To better demonstrate N-terminal Shh processing during release, we inverted and colored (gray) the scale blots obtained from α-Shh antibody (binds processed and unprocessed Shh) and α*-*CW antibody (binds the putative cleavage site) incubation. Yellow signals in merged blots thus denote N-unprocessed proteins bound by both antibodies, and green signals confirm the removal of N- terminal peptides (arrow). Complete processing of N-palmitoylated N-termini (Shh) and incomplete processing of non-palmitoylated termini (ShhC25A) is confirmed by differential α*-*CW-antibody binding. Right: Protein processing converts otherwise inactive ShhC25A into the active morphogen, as indicated by induced Hh-dependent C3H10T_1/2_ reporter cell differentiation into osteoblasts. *** denotes statistical significance (p<0.0001), **denotes statistical significance (p<0.0047). n.s.: not significant (<0.05). Taken from (11).

Finally, peptides cannot mimic the extended conformational Hh binding site, and as has been outlined in our earlier work (14), N-peptides are only loosely associated with the Ptc-binding site of adjacent molecules, so cleavage will efficiently remove them. The suggested experimental approach is therefore not usable, also because authentic multimeric Hh is required for the suggested tests that has never been produced and purified in sufficient amounts.

Altogether, the authors propose a model in which Hh is secreted to the plasma membrane as dually lipidated molecule, assembled as multimer at the cell surface, and then cleaved for propagation. What is surprising with the author's model is that both lipids need to be cleaved to release Hh functional cluster from producing cells. This is intriguing as numerous labs have shown that expression of a HH variant without lipid is secreted as monomere with no activity. So, the authors would need to resolve two important questions to confirm their model. First, how the cleaved HH molecule is kept assembled as multimer, and second, how the putative multimer of non-lipidated HH proposed here is able to provide signaling activation when a monomeric non-lipidated HH cannot.

We note that there is no contradiction: published studies that the reviewer refers to describe HhN^C85S^ that had never been lipidated, as it lacked the C-terminal autoprocessing/cholesterol transferase domain and the palmitate acceptor cysteine.

Such proteins are directly secreted in monomeric (inactive) form, like our protein now shown in Figure 3 of the resubmitted manuscript. Lipidated Hh proteins, in contrast, first associate with the cell surface as a prerequisite to form large multimeric complexes (18), and only then can get proteolytically processed and activated (See Author response image 1). Thus, only Hh produced in lipidated form generates multimers, and proteolytic truncation makes the Ptc-binding site available. This has been published (12-14). Continued association of soluble clusters are best explained by the very large protein/protein interface of about 1000 Angström^2^ (14).

The second question implies that the “putative” multimer (actually, it is firmly established that lipidated Hh multimerizes) cannot provide signalling activation, if the monomer cannot. The explanation is given above: Monomeric HhN^C85S^ had never been lipidated, is soluble, and secreted without processing. Multimeric Hh had been dual- lipidated and – due to removal of inhibitory N-peptides, see above – becomes activated during release. The mechanism is described in detail in published papers and summarized under answer #7.

Reviewer #3:The manuscript by Grobe and colleagues use the Drosophila wing as the principal model system to analyse the functional role of protein palmitoylation in the N-terminal cysteine of Hedgehog (Hh) signalling molecule. Authors present evidence that mutant forms of Drosophila Hh unable to be palmitoylated (Hh-C85S) or proteolytically processed (HA-Hh) gives rise to a Hh-loss of function adult wing phenotype when overexpressed in Hh-expressing but not responding cells, and that truncation of the N-terminal region of Hh-C85S or a mutation of the N-terminal cysteine in HA-Hh restore these phenotypes. Authors propose that palmitoylation of Hh contributes to the clustering of Hh molecules at the cell membrane of Hh-expressing cells and to the tightly controlled-proteolytic processing and release of an active form of Hh able to bind to Ptc in nearby cells.The paper is excellently written and the proposed working model is consistent with the results. Authors have done an excellent and thorough exercise by using the Drosophila wing together with EM data and cell culture experiments and propose a mechanistic model of the biological function of Hh-palmitoylation.Major issue:Authors conclude that the adult wing phenotype caused by expression of Hh-C85S or HA-Hh in the P compartment is caused by a dominant negative effect on the endogenous Hh molecule (as nicely depicted in their cartoons). Authors should present evidence (1) that the wing phenotype can be rescued by co-expression of a wild type form of Hh, (2) that release of a wild type form of Hh into the adjacent compartment is compromised by Hh-C85S or HA-Hh co-expression (by using available UAS-Hh-GFP transgenic flies), and (3) that anterior cells abutting the P compartment are still sensing Hh signalling (similarly to a tethered form of Hh). Thus, expression of a wild type form of Hh and Hh target genes should be analysed in wing primordia of all genotypes (en-gal4; UAS-Hh-C85S +/- UAS-Hh-GFP and en-gal4; UAS-Hh-A +/- UAS-Hh-GFP) to confirm their proposal. These experiments are feasible.

We thank this reviewer for the very useful suggestion to conduct rescue experiments. The obtained results support our model, and are thus included in the re-submitted manuscript (Figures 9-10).

1) We now show that, under en-control, ^HA^Hh^C85S^-induced wing mispatterning is rescued by transgenic Hh. These experiments confirm that “diluting” inhibitory peptides (by increasing relative amounts of Hh) compensates for the C85S mutation.

In notable contrast, en>^HA^Hh leads to pharate lethality, and Hh co-expression did not rescue this phenotype (only very few flies hatched, these showed severe wing mispatterning). This is because ^HA^Hh (even if “diluted”) still retains associated Hh at the surface of producing cells, confirming our model.

2) To show that release of a wild type form of Hh into the adjacent compartment is compromised by ^HA^Hh^C85S^ or ^HA^Hh co-expression, we performed the same rescue experiment in the absence of any endogenous Hh expression in the anterior compartment. It is well-established that anterior Hh misexpression (caused by a natural mutation called *moonrat*) causes anterior wing overgrowth. Forced Hh expression in anterior cells under 34B-Gal4 control results in the same phenotype. The expression of inactive Hh^C85S^ and HAHh alone do not cause *moonrat* phenotypes, as expected. However, 34B>Hh induced anterior wing overgrowth is completely reversed by HAHh (again by retaining Hh at the surface of producing cells), but Hh^C85S^ merely reduced the *moonrat* gain-of-function phenotype (by incomplete blockade of Hh Ptc-binding sites).

3) En/ptc/dpp target gene expression at the A/P border is consistent with the obtained phenotypes.

Please note that, for these experiments, we refrained from using suggested Hh- GFP constructs (2) for the reasons given above to reviewer 1 and 2. In our hands, tagged Hh^C85S^-GFP (a) fails to inhibit endogenous Hh function if expressed in the same compartment, indicating impaired function due to the large tag, and (b) undergoes extensive proteolytic processing in larvae. In our hands, Hh-GFP also has a much lower activity if compared to untagged Hh. We thus decided to modify the suggested experimental outline and included the results in the manuscript (Figures 9,10)

Other issues:1) Were male of female wings used in their L3-L4/L2-L3 region measurements?

10 male and 10 female (single) wings were analyzed per data set. This is now stated in the Materials and methods section. No differences in wing patterning were observed between the sexes.

2) Authors should notice that en-gal4 is also expressed in the A compartment, although at lower levels and later in development. The ability of Hh-C85S and HA-Hh to cause a wing phenotype should be analysed also with the hh-gal4 driver (again available in the fly community).

Hh-Gal4 (Bloomington #67046, FlyBaseID FBti0017278) fully reproduced our results, as now shown in Supplement 4. Another en-line (Bloomington #48011) also reproduces our observed phenotypes. Data are now included (Figure 4—figure supplement 5).

3) Expression of Hh-C85S in the ptc domain has not phenotype (Figure 3H) while Hh-HA does Figure 6H, I). Why?

As stated, the intervein area was about 7% smaller upon ptc- mediated Hh^C85S^ expression, so a small effect was present. Stronger phenotypes were observed for some truncated C85S constructs (see Author response image 2). One idea to explain this is that transgenic proteins may saturate Heparan sulfate binding sites in the anterior field, in turn somehow reducing transport or activity of endogenous Hh.

**Author response image 2. respfig2:** Compared phenotypes resulting from the overexpression of Hh^C85S^ and N- terminally truncated variants in the Hh-producing and Hh-receiving compartments of the wing disc. (**A-F**) Adult wing structures as a result of *ptc-Gal4*-mediated overexpression of non- palmitoylated Hh in wild type background (at 25 °C). Note that proximal L3-L4 intervein spaces get slightly reduced (**B-D**) and, in addition, varying numbers of wings lacked the ACV (E and Table 1 below). Wing patterning is fully restored upon deletion of the most N-terminal 15 amino acids.

4) How certain are authors that veins in Figure 6B are L4 and L5 and not L2 and L5. The latter would be expected as these are the ones that depend on Dpp signalling.

We agree with this reviewer that veins shown in this figure cannot be clearly assigned. This has been changed in the revised MS.

References:

1. D'Angelo G, Matusek T, Pizette S, Therond PP. Endocytosis of Hedgehog through dispatched regulates long-range signaling. Dev Cell. 2015;32(3):290-303.

2. Torroja C, Gorfinkiel N, Guerrero I. Patched controls the Hedgehog gradient by endocytosis in a dynamin-dependent manner, but this internalization does not play a major role in signal transduction. Development. 2004;131(10):2395-408.

3. Callejo A, Culi J, Guerrero I. Patched, the receptor of Hedgehog, is a lipoprotein receptor. Proc Natl Acad Sci U S A. 2008;105(3):912-7.

4. Callejo A, Bilioni A, Mollica E, Gorfinkiel N, Andres G, Ibanez C, et al. Dispatched mediates Hedgehog basolateral release to form the long-range morphogenetic gradient in the *Drosophila* wing disk epithelium. Proc Natl Acad Sci U S A. 2011;108(31):12591-8.

5. Bischoff M, Gradilla AC, Seijo I, Andres G, Rodriguez-Navas C, Gonzalez-Mendez L, et al. Cytonemes are required for the establishment of a normal Hedgehog morphogen gradient in *Drosophila* epithelia. Nat Cell Biol. 2013;15(11):1269-81.

6. Gradilla AC, Gonzalez E, Seijo I, Andres G, Bischoff M, Gonzalez-Mendez L, et al. Exosomes as Hedgehog carriers in cytoneme-mediated transport and secretion. Nat Commun. 2014;5:5649.

7. Vincent S, Thomas A, Brasher B, Benson JD. Targeting of proteins to membranes through hedgehog auto-processing. Nat Biotechnol. 2003;21(8):936-40.

8. Palm W, Swierczynska MM, Kumari V, Ehrhart-Bornstein M, Bornstein SR, Eaton S. Secretion and signaling activities of lipoprotein-associated hedgehog and non-sterolmodified

hedgehog in flies and mammals. PLoS Biol. 2013;11(3):e1001505.

9. Haines N, van den Heuvel M. A directed mutagenesis screen in *Drosophila melanogaster* reveals new mutants that influence hedgehog signaling. Genetics. 2000;156(4):1777-85.

10. Zhou S, Lo WC, Suhalim JL, Digman MA, Gratton E, Nie Q, et al. Free extracellular diffusion creates the Dpp morphogen gradient of the *Drosophila* wing disc. Curr Biol. 2012;22(8):668-75.

11. Jakobs P, Schulz P, Schurmann S, Niland S, Exner S, Rebollido-Rios R, et al. Calcium coordination controls sonic hedgehog structure and Scube2-cubulin domain regulated release. J Cell Sci. 2017.

12. Jakobs P, Schulz P, Ortmann C, Schurmann S, Exner S, Rebollido-Rios R, et al. Bridging the gap: heparan sulfate and Scube2 assemble Sonic hedgehog release complexes at the surface of producing cells. Sci Rep. 2016;6:26435.

13. Jakobs P, Exner S, Schurmann S, Pickhinke U, Bandari S, Ortmann C, et al. Scube2 enhances proteolytic Shh processing from the surface of Shh-producing cells. J Cell Sci. 2014;127(Pt 8):1726-37.

14. Ohlig S, Farshi P, Pickhinke U, van den Boom J, Hoing S, Jakuschev S, et al. Sonic hedgehog shedding results in functional activation of the solubilized protein. Dev Cell. 2011;20(6):764-74.

15. Ohlig S, Pickhinke U, Sirko S, Bandari S, Hoffmann D, Dreier R, et al. An emerging role of sonic hedgehog shedding as a modulator of heparan sulfate interactions. J Biol Chem. 2012;287(52):43708-19.

16. Dierker T, Dreier R, Migone M, Hamer S, Grobe K. Heparan sulfate and transglutaminase activity are required for the formation of covalently cross-linked hedgehog oligomers. J Biol Chem. 2009;284(47):32562-71.

17. Pepinsky RB, Rayhorn P, Day ES, Dergay A, Williams KP, Galdes A, et al. Mapping sonic hedgehog-receptor interactions by steric interference. J Biol Chem. 2000;275(15):10995-1001.

18. Vyas N, Goswami D, Manonmani A, Sharma P, Ranganath HA, VijayRaghavan K, et al. Nanoscale organization of hedgehog is essential for long-range signaling. Cell. 2008;133(7):1214-27.

[Editors' note: the author responses to the re-review follow.]

The manuscript has been improved but there are some remaining issues that need to be addressed before acceptance, as outlined below.The editors and reviewers have discussed your paper extensively. Our position is that your work is interesting, provocative and of potential broad significance. On the other hand, there is a clear consensus that you do not currently provide a clear enough description of the controversy and uncertainty that surrounds your interpretation. Your paper does not yet allow a definitive conclusion about the model.eLife is an appropriate journal for this kind of work, but only if the authors are able clearly and explicitly to describe their work in the wider context, allowing a reader to form a balanced picture of where the field stands. We would therefore like to give you one more opportunity to respond to remaining concerns but also – and even more importantly – to rewrite the Introduction and Discussion in a way that gives that broader context and describes to a non-expert reader the uncertainties that remain about your model. What would be a definitive experiment that might prove or refute it? Which areas are you least confident about? Do you have a plan about how to address them? How do you reconcile your ideas with previous apparently contradictory data?Science is full of uncertainty and journals like eLife should not encourage this to be hidden behind unrealistic confidence or advocacy. If you can amend your paper to provide a fair and balanced description of how the field stands, and how your new data move it forward, but also where the uncertainties still lie, we would be happy to publish it. If you feel that this is not in your best interest and you prefer to argue for your model more directly, our decision would be not to accept it.

We are grateful for the opportunity to revise our manuscript “Proteolytic processing of palmitoylated Hedgehog peptides specifies the 3-4 intervein region of the *Drosophila* wing” according to your suggestions and the final points raised by the reviewers. Following the line of a recently published paper in *eLife*, “Cytoneme-mediated cell-cell contacts for Hedgehog reception” by Gonzales-Mendez et al., we agree that our findings can – and should – be related to previous findings on Hh morphogen release and transport to provide a more balanced view of where the field currently stands. We also added a balanced view on the strengths and weaknesses of our concept and of ways to test them. Specifically, we changed the following parts of our manuscript and marked all changes in red in the related manuscript version called “Schuermann et al. marked changes”.

1) In the Discussion section, we added a paragraph describing the strengths and weaknesses of our concept of proteolytic Hh release, and we suggest experiments required to support our concept in other tissues. The greatest uncertainty we see in our concept is that N-truncation of non-palmitoylated Hh^C85S^ rescues wild-type Hh activity in the wing disc, but does not lead to a gain-of-function (in contrast to the situation in cell culture, as described below). A possible experiment to test this problem is now discussed and (after we had been planning this long before) will be started in the lab.

2) We also discuss potential congruencies between our concept of proteolytic Hh release and activation and the concepts of others, specifically, the possibility that Hh shedding relays Hh at the cytoneme synapse. To this end, we added Figure 12 to illustrate and describe the various concepts of membrane-dependent and -independent Hh transport.

3) We reworded the Introduction and Discussion to make our results and conclusions less definitive, as requested by the editors.

In addition to these general points there are still some remaining specific issues in the reviews below. We would encourage you to address these as fully as possible. Where necessary and appropriate, please acknowledge if they provide significant challenges to your model.Reviewer #1:[…] Overall, this is a story that is ultimately still not convincing for this reviewer. The model proposed here cannot explain many of the in vivo observations obtained by other labs, some of which are listed above.

Reviewer 1 stated that a paper by Lee et al. (2001) demonstrates that non-palmitoylated Shh variants expressed in the posterior region of the mouse limb have no inhibitory effect on Hh patterning. We were aware of this finding and the fact that ectopic overexpression of non-palmitoylated Shh in the mouse even induces significant gain-of-function phenotypes (Chen et al., 2004; Kohtz et al., 2001; Lee et al., 2001) and that loss of palmitoylase activity in Hhat mutants causes defects that are characteristic of defective Shh signaling, but are less severe than Shh loss-of-function phenotypes (Chen et al., 2004). However, we believe that our model of coupled N-terminal Hh processing and activation does not contradict these data, as claimed by the reviewer, but rather explains them. Non-palmitoylated full-length and inactive Shh^C25S^ can be converted into bioactive proteins in vitro simply by their N-terminal processing (Jakobs et al., 2017; Ohlig et al., 2012) (Figure 1). A key facilitator of this conversion is the Shh release factor Scube2 (See supplement of Jakobs et al., 2017). Scube2-assisted N-terminal cleavage could thus help convert dominant negative Shh^C25S^ activity in vivo (by blockade of adjacent Ptc-binding sites) into a gain-of-function-phenotype. Notably, *Drosophila* has no Scube orthologs, which may explain persistent Hh^C85S^ dominant negative activity in the fly. Because these data and our interpretation have already been published by our group (Jakobs et al., PMID: 28778988), and because we extensively discussed them in a recently accepted review (Manikowski et al., http://www.mdpi.com/2221-3759/6/1/3/pdf), we prefer not to discuss them again in this manuscript.

We prefer not to discuss several points (mostly raised by reviewer 1) in our revised manuscript for the following reasons:

1) Reviewer 1 used Figure 4 of Lee et al. (2001) to demonstrate a weak inhibitory effect of anterior overexpression of non-palmitoylated Hh variants in the Drosophila wing. We explain the slight dominant negative ptc-Gal4, Hh^C85S^ phenotype by the different integration sites used for transgene generation and the different ptc-driver line used. Yet, exactly as shown in our work, the ptc-Gal4 Hh^C85S^ expression phenotype was much weaker than the phenotypes obtained from posterior HhC85S expression, or from *omb*-controlled expression at the a/p border. We thus believe that discussing the isolated ptc-Gal4;Hh^C85S^ result out of its experimental context would be misleading.

2) A second criticism raised by the reviewer is based on a publication of Williams et al. (1999) that does not show stoichiometric suppression of wild-type protein function by non-palmitoylated Hh. We note that the Hhs analyzed in this study were monomeric variants intracellularly expressed in (and purified from) yeast (or *E. coli*). This system thus does not allow for physiological Hh multimerization at the surface of expressing cells and inhibitory interactions *in trans*, as required by our model. Therefore, we see no contradiction.

3) The reviewer also refers to Figure 7 in Chen et al. (2004), showing size exclusion results obtained from various expressed Hh, Shh, and variants lacking one or both lipidations. Hh^C85S^ forms multimers, consistent with the results shown in our manuscript (Figure 1F, Figure 4—figure supplements 1 and 2, Figure 6—figure supplement 1) – and not contradicting them, as claimed. The linear Hh structure is also not extrapolated but supported by a) established Hh multimerization using linear cell-surface HS chains as scaffolds, and b) our IEM data (Figure 1—figure supplement 1). One side note: Most dual-lipidated Hh, Shh, and monolipidated Hh^C85S^ and Shh^C25S^ proteins analyzed by Chen et al. (2004, Figure 7) were monomeric, probably due to the extended (2-day) serum-free expression conditions used – this result should therefore be interpreted with care.

4) Williams et al. (1999) were again cited to suggest that N-truncated proteins are inactive, supposedly at odds with our model. Again, we note that only non-physiological *Pichia*- or *E. coli*-expressed monomeric proteins were used in this study. Still, the same paper shows that some N-truncated proteins were bioactive in some in vitro systems, and a non-palmitoylated version lacking 9 N-terminal amino acids was strongly active in vivo (in Ptc-reporter mice, Figure 7 in the work of Williams et al). This demonstrates that in Ptc-reporter mice a) N-palmitate is not directly required for Hh biofunction, consistent with our data, and b) N-truncated non-palmitoylated Hh is bioactive, again consistent with our model.

5) It was further claimed that full-length unlipidated Hh^C85S^ is active, as supposedly shown in two reports (Gallet et al., 2003, Figure 2, and Callejo et al., 2006, Figure 6). If correct, this would be in contrast with our data and with reports of others (Chamoun et al., 2001; Lee et al., 2001). Gallet et al. show that Hh^C85S^ retains signaling activity in a different in vivo system and by using the ubiquitous 69BGal4 driver. Still, if expressed under en-control at physiologically relevant sites, the potency of Hh^C85S^ to activate *rho*, repress *ser*, and upregulate *ptc* is reduced, as also stated in their figure legend and consistent with our findings. The authors further show that an artificial, permanently membrane-tethered Hh form (Hh-CD2) is completely inactive, consistent with Strigini and Cohen (1997) and ^HA^Hh results shown and discussed in our manuscript.

Figure 6 by Callajo et al. (2006) shows that Hh^C85S^, unlike Hh, fails to induce high-threshold *ptc*- and *en*-expression inside and outside of anterior clones, which is consistent with our findings. *Dpp* and *iro* expression are not expanded but strongly restricted. These two results are very different from the reviewer’s claim that “One should note that overexpression of this variant has been published previously by several other groups and showed an opposite result, with expansion of target genes in both embryos and wing imaginal discs.”

6) Finally, the reviewer cannot reconcile reported bioactivity of variably N-terminally lipidated Hhs (or Hhs carrying terminal isoleucines) with our model. Our model predicts that the exact nature of the Hh lipid is not crucial, because palmitate can be potentially replaced by any hydrophobic moiety as long as it tethers the protein to the membrane and does not interfere with shedding. Taylor et al. (2001, but also others: Long et al., 2015) show exactly that: As long as the Hh N-terminus is hydrophobic and the cleavage site accessible (unlike for Hh-CD2 and ^HA^Hh), the protein is functional. This does not contradict our model but is in line with it (in contrast to the models of others, claiming that palmitate signals directly to the Ptc receptor (Tukachinsky et al., 2016)).

Reviewer #2:Schuermann et al. provide evidence that Hh activity and spread depend on the proteolytic removal of the palmitoylated N-terminal peptide. The paper contains a huge amount of data but there is no "killer experiment" that definitively proves the proposed model. For example, most of the conclusions rest on the use of ectopic expression of mutant versions in a wild-type background instead of replacement of the endogenous hh gene. Moreover, no in vivo localization data is provided. I therefore encourage the authors to raise the possibility that there might be other interpretations of the data and discuss the need for follow up studies of Hh versions expressed from the endogenous locus and visualized in vivo. Despite these concerns, the findings are interesting, provocative and of broad interest.

Reviewer 2 suggests a need for follow-up studies of Hh versions expressed from the endogenous locus and visualized in vivo. We will try this experiment in the future. We already tried Hh visualization in the Ptc-internalization-deficient ptc^28^TPT fly line from Gary Struhls lab; still, at least in our hands, direct Hh stainings in the wing disc were unsatisfactory.

Other concerns:1) Is the packing seen in Figure 1 really due to Hh or could it be caused by the confirmation and size of the antibodies?

Reviewer 2 suggests that the packing seen in Figure 1 might be due to the conformation and size of the antibodies. In the original manuscript and the revised version, Figure 1—figure supplement 1 shows similar data obtained from three different antibodies alone and in combination. To the best of our knowledge, the observed conformations are therefore not a consequence of unspecific antibody clustering.

2) The evidence that the mutant versions are expressed at similar levels as wild-type Hh is sparse.

We would like to reply that all constructs are present at the same insertion site and that their comparable expression was confirmed in vitro. In vivo confirmation of comparable expression was tried bud did not yield useful results due to excessive background and unspecific protein staining on Westerns. Transgene expression from endogenous sites, as suggested by the reviewer, is certainly a very good idea and will be performed in the future.